# Focusing attention in working and long-term memory through dissociable mechanisms

Dongyu Gong [1,2,3] ✉, Dejan Draschkow [1,2] ✉ & Anna C. Nobre [1,2,3,4] ✉

We developed an experimental approach to compare how attentional orienting facilitates retrieval from spatial working memory (WM) and long-term memory (LTM), and how selective attention within these two memory types impacts incoming sensory information processing. In three experiments with healthy young adults, retrospective attention cues prioritize an item represented in WM or LTM. Participants then retrieve a memory item or perform a perceptual task. The retrocue is informative for the retrieval task but not for the perceptual task. We show that attentional orienting benefits performance for both WM and LTM, with stronger effects for WM. Eye-tracking reveals significant gaze shifts and microsaccades correlated with attention in WM, but no statistically significant gaze biases were found for LTM. Visual discrimination of unrelated visual stimuli is consistently improved for items matching attended WM locations. Similar effects occur at LTM locations but less consistently. The findings suggest at least partly dissociable attention-orienting processes for different memory types. Although our conclusions are necessarily constrained to the type of WM and LTM representations relevant to our task, they suggest that, under certain conditions, attentional prioritization in LTM can operate independently from WM. Future research should explore whether similar dissociations extend to non-spatial or more complex forms of LTM.

Selective attention is fundamental to human cognition, delivering information that is relevant to ongoing behaviour among boundless competing signals[1–4]. While standard models emphasise external sensory inputs as a primary target of attentional control[1,5], the arrow of attention can also point inward[6,7] to flexibly prioritise and select items within working memory (WM). Retrospective cues presented during WM delay period that predict goal-relevant items have proved highly useful for investigating the modulatory mechanisms of internal selective attention[8–10].

Interestingly, like during attentional selection in the external world[11–13], orienting attention within visual WM is accompanied by subtle gaze shifts[14–16]. Studies have also shown that selecting an item in WM leads to the capture of attention by sensory stimuli with matching locations or features[17–19]. These capture effects appear obligatory, operating even when they are counter-productive for the perceptual task[20], thus underscoring the strong role of WM in guiding external attention.

While there is a solid appreciation for the strong interrelation between attention and WM, a similar symbiotic relationship has been proposed for attention and long-term memory (LTM). LTM has long been considered to guide sensory processing, as exemplified by Helmholtz's view that perception results from unconscious inference based on prior memories[21]. Experimental tasks manipulating prior experience with stimulus contexts have since confirmed the role LTM plays in guiding attention[22–28].

In turn, researchers have posited that selective attention also operates within LTM, guiding retrieval. Corroborating evidence comes from the observation that brain areas controlling external attentional

[1]Department of Experimental Psychology, University of Oxford, Oxford, UK. [2]Oxford Centre for Human Brain Activity, Wellcome Centre for Integrative Neuroimaging, Department of Psychiatry, University of Oxford, Oxford, UK. [3]Department of Psychology, Yale University, New Haven, CT, USA. [4]Wu Tsai Institute, Yale University, New Haven, CT, USA. ✉e-mail: dongyu.gong@yale.edu; dejan.draschkow@psy.ox.ac.uk; kia.nobre@yale.edu

orienting are activated in tasks of episodic memory retrieval[29–32] (but see ref. [33]). Studies showing similar modulation of alpha-band neural activity during visual memory retrieval as during external visual attention further support a role for attentional selection in guiding LTM retrieval[34,35]. However, less is understood about how selective attention prioritises different contents within LTM. One major open question is whether the ability to focus attention on specific contents within LTM, to guide selective retrieval, relies on the same internal attention processes that operate in WM. Most standard models of memory suggest that LTM retrieval is mediated by WM[36,37]. If representations move to WM, then the same internal attention mechanisms should select relevant mnemonic content for behaviour. However, in principle, focusing on LTM content to guide retrieval may also occur directly and independently.

To test how selective attention modulates LTM retrieval, we need experimental tasks that manipulate the goal relevance of items in LTM. Such tasks should provide direct evidence that orienting attention to a specific item in LTM yields performance benefits relative to retrieving other items associated with the same context. With evidence suggesting the ability to orient attention in LTM flexibly and selectively, it becomes possible to probe how similar the processes and consequences are compared to orienting in WM. Would orienting in LTM also be accompanied by gaze biases? Would the selective attention in LTM spill over to modulate sensory processing? Answering these questions would inform enduring debates regarding the relationship between WM and LTM[36,38–41].

Here, we introduce an experimental framework to test the ability to orient internal spatial attention selectively among competing LTM contents and to compare directly selective attention in WM and LTM. We borrow from retrospective cue (referred to as "retrocue" hereafter) designs to direct attention to a specific item within a pre-learned array of items in LTM or to a specific item of an encoded WM array presented earlier in the trial. Across three experiments, we measured the consequences of orienting spatial attention in WM vs. LTM for memory retrieval and additionally tested for spill-over effects on sensory processing. At the end of each trial, participants either retrieved a memory item or performed a perceptual discrimination task. The retrocue was informative for the retrieval task but not for the perceptual task. This allowed us to investigate how orienting selective attention in WM and LTM enhanced the retrieval of relevant memoranda and biased perceptual processing in an unrelated task in similar ways.

The findings demonstrate the significant effects of internal attention in WM and LTM and, interestingly, show that dissociable mechanisms are at play. Orienting attention in LTM consistently speeds selective retrieval and only sometimes improves retrieval accuracy. Benefits of internal attention in WM are stronger, with larger gains for response speed and consistent accuracy improvements. Eye-tracking reveals notable differences in engagement of oculomotor mechanisms during shifts of attention within WM vs. LTM domains. Perceptual discrimination of unrelated visual stimuli is consistently superior at locations associated with LTM items and is modulated by internal attention in both WM and LTM domains.

## Results

Across three experiments, we demonstrate benefits conferred by orienting attention in WM and LTM, which differed qualitatively. In experiment 1 (Figs. 1 and 2), participants acquired two colour-location bindings in LTM during a training session. In a subsequent testing session, they encoded two additional colour-location bindings into WM on every trial. Half of the trials tested for the benefits of orienting attention in WM or LTM. Participants indicated the location of an item based on informative or non-informative colour retrocues. The remaining half tested the impact of prioritising an item location in WM or LTM on sensory processing. Participants discriminated the direction

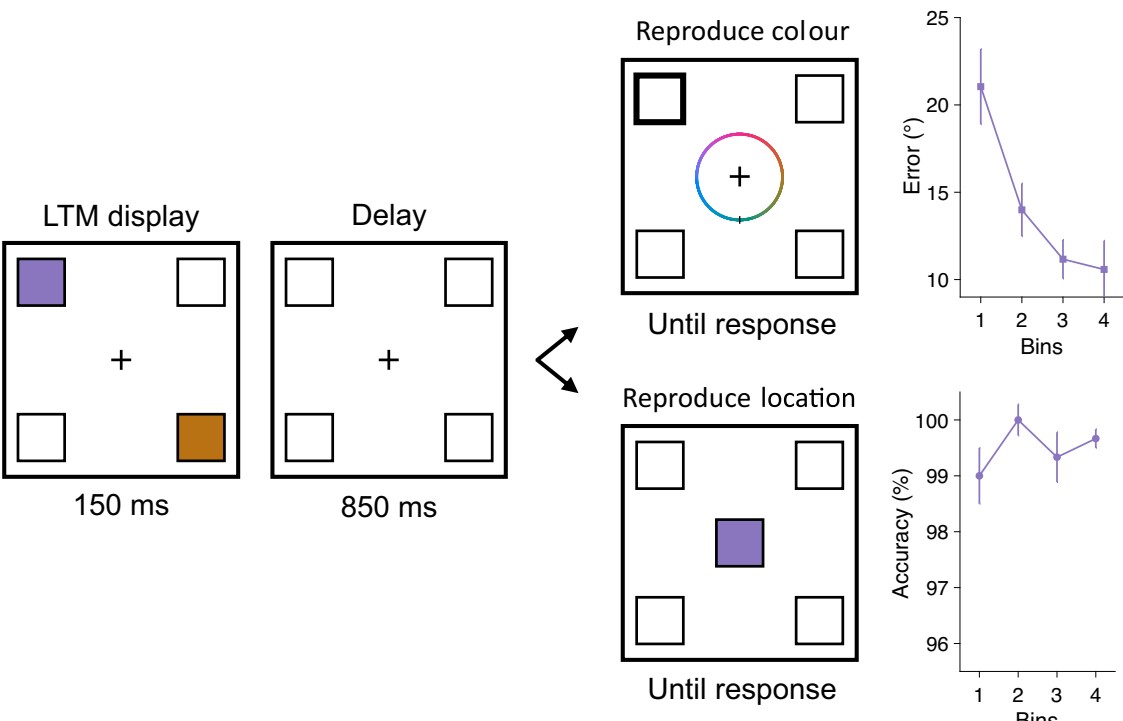

**Fig. 1 | LTM training in experiment 1.** On every learning trial, participants (*n* = 30) memorised the same colours and locations of two squares at diagonal locations. When prompted, they reproduced the colour at the probed location or the location of the probed colour. Colour reproduction trials and location reproduction trials were presented in random order. Learning performance was aggregated into four bins (of 10 trials each) and shown in the form of errors for colour reproduction and accuracy for location reproduction, respectively. Error bars represent ±1 SEM.

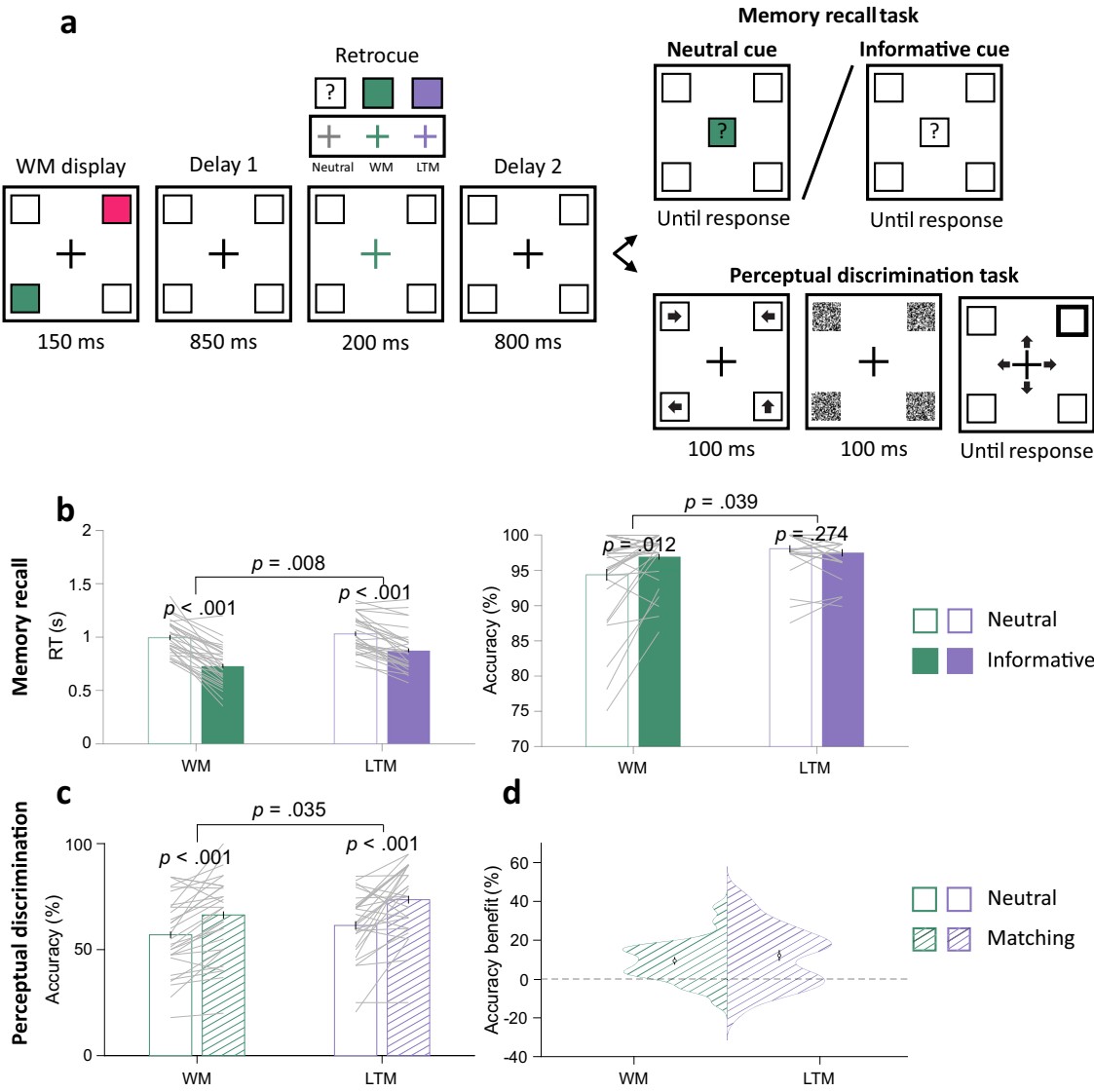

**Fig. 2 | Behavioural results in experiment 1. a** Task schematic of the testing trials in experiment 1. Participants first encoded two items into WM, with different colours and locations from those in LTM. After the first delay, neutral retrocues were uninformative, whereas informative retrocues (WM or LTM) indicated which location would be relevant in the memory retrieval task. After a second delay, the memory retrieval task and the perceptual discrimination task were equally likely to be presented. In the retrieval task, participants retrieved the location of a randomly chosen colour on neutral-retrocue trials and the location of the retrocued colour on informative-retrocue trials. In the perceptual task, participants discriminated the direction of one of the four arrows that briefly appeared at the four quadrants before being masked. The location of the probed arrow was indicated by a post-cue. **b** Memory retrieval mean RTs and accuracy for WM and LTM items, grouped by neutral-retrocue and informative-retrocue trials. **c** Perceptual discrimination accuracy at WM and LTM locations, grouped by retrocue neutral and matching trials. **d** Means and distributions of perceptual accuracy benefits for WM and LTM locations matching retrocues. Error bars in **b**–**d** represent ±1 *SEM* across participants ($n = 30$). In **b**, **c**, *p* values are reported according to two-way ANOVAs or post-hoc *t* tests with Bonferroni corrections.

of one of four arrows briefly flashed at WM and LTM locations. Experiments 2 (Fig. 3 and 4) and 3 (Fig. 5) followed the same training-testing approach, with testing trials measuring benefits of internal attention on memory performance or sensory processing. Experiment 2 extended the findings to colour-shape bindings. A non-spatial task probed for benefits of internal attention. Participants reproduced item shapes using a randomly oriented wheel after informative or non-informative colour retrocues. This shape reproduction method controlled for the possible contributions of motor preparation in memory retrieval performance. The perceptual discrimination trials were the same as in experiment 1. Experiment 3 was based on experiment 2, with training and testing sessions separated across two successive days to ensure that LTM effects did not reflect lingering priming or shorter-term memory traces. The perceptual discrimination task was replaced with a simpler task. Only one arrow appeared, and participants identified its direction.

## Training LTM associations

For each experiment, we first confirmed that our LTM training was effective. In experiment 1 (Fig. 1), a linear-contrast analysis showed that participants reproduced the LTM colours with smaller errors on late as compared to early trial bins ($F(1, 26) = 16.156$, $p < 0.001$, partial $\eta^2 = 0.157$). For the location reproduction task, participants remained near ceiling-level accuracy across the four bins (first bin: $0.990 \pm 0.005$ ($M \pm$ SEM), last bin: $0.997 \pm 0.002$, $F(1, 26) = 0.480$, $p = 0.490$). In experiment 2 (Fig. 3), shape reproduction improved significantly across bins (linear contrast: $F(1, 40) = 17.939$, $p < 0.001$, partial $\eta^2 = 0.122$). Although improvement in colour reproduction was not

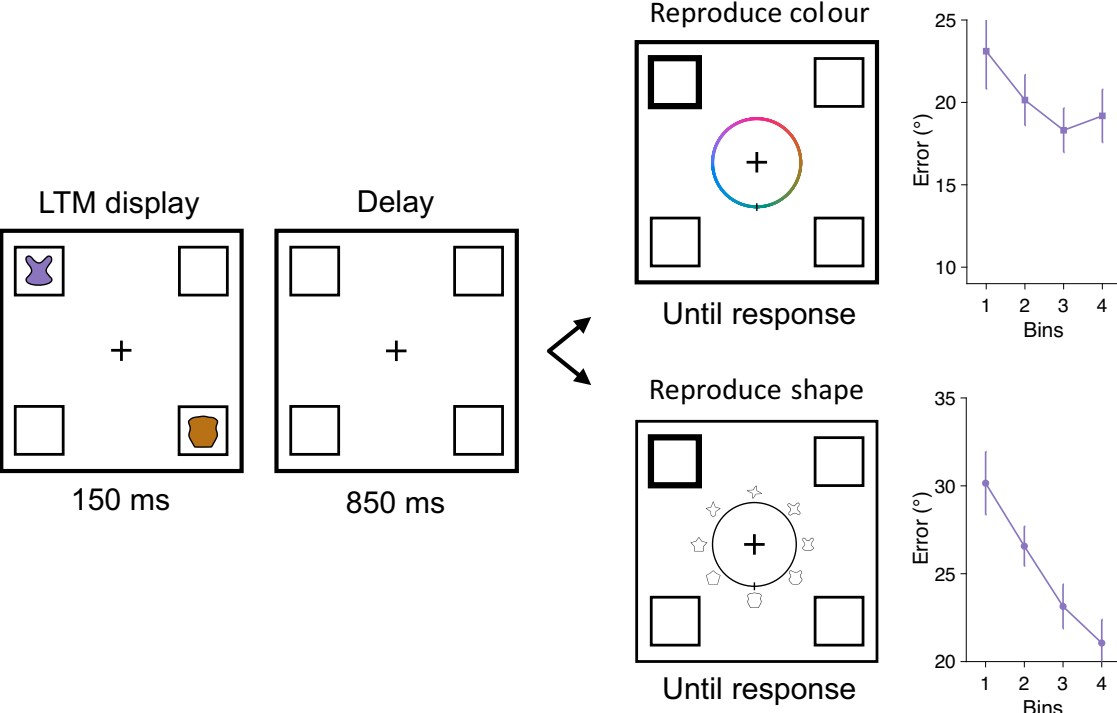

**Fig. 3 | LTM training in experiment 2.** On every learning trial, participants (*n* = 44) memorised two consistent items at diagonal locations defined by unique colours and shapes. When prompted, they reproduced the colour or shape at one location. Colour reproduction trials and shape reproduction trials appeared in random order. Learning performance was sorted into 4 bins (with 10 trials each) and shown in the form of errors for both colour and shape reproduction. Error bars represent ±1 *SEM*. Shape stimuli are adapted from Li, A. Y., Liang, J. C., Lee, A. C. H., & Barense, M. D. (2020). The validated circular shape space: Quantifying the visual similarity of shape. *Journal of Experimental Psychology*: General, 149, 949–966. https://doi.org/10.1037/xge0000693.

significant ($F(1, 40) = 2.304$, $p = 0.132$), the errors remained relatively low with a numerically decreasing trend across bins (first bin: 23.107° ± 2.295, last bin: 19.182° ± 1.612). In experiment 3 (Fig. 5b), linear contrasts showed that learning performance improved significantly for both colour reproduction ($F(1, 115) = 9.015$, $p < 0.001$, partial $\eta^2 = 0.054$) and shape reproduction ($F(1, 115) = 38.470$, $p < 0.001$, partial $\eta^2 = 0.196$).

**Attentional orienting exerts stronger effects in WM than LTM**
Across three experiments, attentional orienting exerted stronger effects in WM than LTM, featuring higher benefits in both retrieval speed and accuracy.

In experiment 1 (Fig. 2b), both retrocueing (neutral vs. informative) and memory conditions (WM vs. LTM) significantly impacted retrieval speed: RTs in the memory retrieval task were faster when retrocues were informative ($F(1, 29) = 83.205$, $p < 0.001$, partial $\eta^2 = 0.742$) and when WM items were probed ($F(1, 29) = 8.239$, $p = 0.008$, partial $\eta^2 = 0.221$). Crucially, retrocueing and memory conditions also interacted ($F(1, 29) = 18.627$, $p < 0.001$, partial $\eta^2 = 0.391$). Both WM and LTM retrocues conferred a significant benefit (WM: $t(29) = 9.217$, $p_{Bonferroni} < 0.001$, $d = 1.521$; LTM: $t(29) = 6.575$, $p_{Bonferroni} < 0.001$, $d = 0.833$), but retrocue benefits were stronger for the speed of retrieving WM items ($t(29) = 4.316$, $p < 0.001$, $d = 0.748$). Similar analyses on retrieval quality showed significant main effects of retrocueing ($F(1, 29) = 4.340$ $p = 0.046$, partial $\eta^2 = 0.130$) and memory conditions ($F(1, 29) = 4.667$, $p = 0.039$, partial $\eta^2 = 0.139$), as well as a significant interaction ($F(1, 29) = 12.115$, $p = 0.002$, partial $\eta^2 = 0.295$). Post-hoc comparisons revealed a significant improvement in retrieval accuracy by retrocues for WM items ($t(29) = 2.981$, $p_{Bonferroni} = 0.012$, $d = 0.465$) but no significant effect of retrocues for LTM items ($t(29) = 1.530$, $p_{Bonferroni} = 0.274$). This could possibly be explained by the overall very high retrieval accuracy for LTM (0.977 ± 0.005),

leaving little room for improvement. A shortcoming of experiment 1 was the spatial nature of the memory-retrieval report. Upon presentation of an informative retrocue, participants could immediately prepare their response, thus confounding the quality and speed of item selection with response preparation. The non-spatial nature of experiments 2 and 3, combined with using a randomly oriented response wheel, overcame this limitation.

In experiment 2 (Fig. 4b), retrocueing significantly shortened RTs ($F(1, 43) = 179.198$, $p < 0.001$, partial $\eta^2 = 0.807$). Although RTs were equivalent for both memory conditions ($F(1, 43) = 0.436$, $p = 0.513$), there was a significant interaction between the two factors ($F(1, 43) = 24.568$, $p < 0.001$, partial $\eta^2 = 0.364$). RT benefits were present for both WM ($t(43) = 11.657$, $p_{Bonferroni} < 0.001$, $d = 0.594$) and LTM ($t(43) = 9.016$, $p_{Bonferroni} < 0.001$, $d = 0.273$) but were stronger for WM ($t(43) = 4.957$, $p < 0.001$, $d = 0.929$). For retrieval accuracy, both retrocueing ($F(1, 43) = 8.971$, $p = 0.005$, partial $\eta^2 = 0.173$) and memory conditions ($F(1, 43) = 32.995$, $p < 0.001$, partial $\eta^2 = 0.434$) exerted significant main effects. The two factors also interacted ($F(1, 43) = 7.441$, $p = 0.006$, partial $\eta^2 = 0.163$). Post-hoc comparisons revealed that WM retrocues significantly reduced shape reproduction errors ($t(43) = 4.322$, $p_{Bonferroni} < 0.001$, $d = 0.429$) but LTM retrocues did not ($t(43) = 1.250$, $p_{Bonferroni} = 0.436$). Shape reproduction was very accurate, and errors were smaller when LTM shapes were reproduced (16.610° ± 1.519) as compared to WM shapes (27.603° ± 2.417), possibly reflecting ceiling effects.

In experiment 3 (Fig. 5c), the one-day interval between training and testing brought LTM performance below ceiling, allowing us to test for LTM accuracy benefits by internal attention. Overall, the next-day LTM representations in experiment 3 were weaker than the same-day representations in experiment 2 (reproduction errors in experiment 3 vs. Experiment 2: 51.547 ± 2.880 vs. 17.378 ± 1.819, Welch's $t = 10.031$, $p < 0.001$). Informative retrocues significantly improved RTs

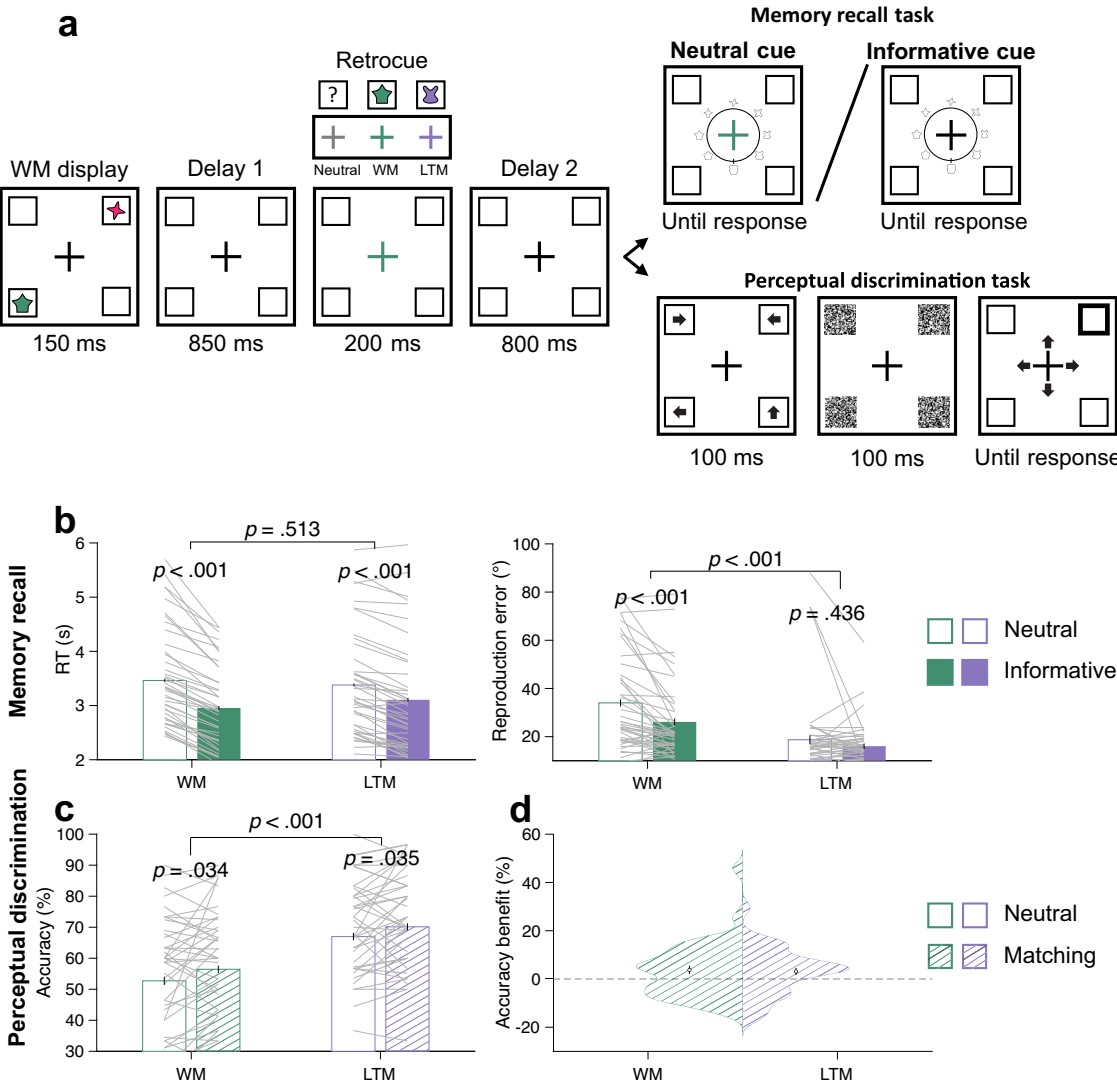

**Fig. 4 | Behavioural results in experiment 2. a** Task schematic of the testing trials in experiment 2. Participants first encoded two items in WM. Their colours, shapes and locations differed from those in LTM. After the first delay, neutral retrocues were uninformative, whereas informative retrocues (WM or LTM) indicated which item shape would be relevant in the memory retrieval task. After a second delay, the memory retrieval task and the perceptual discrimination task were equally likely to be presented. In the retrieval task, participants retrieved the shape of the item matching a randomly chosen colour on neutral-retrocue trials and the shape of the item matching the retrocued colour on informative-retrocue trials. In the perceptual task, participants discriminated the direction of one of the four arrows that briefly appeared at the four quadrants before being masked. The location of the probed arrow was indicated by a post-cue. **b** Memory retrieval mean RTs and accuracy for WM and LTM items, grouped by neutral-retrocue and informative-retrocue trials. **c** Perceptual discrimination accuracy at WM and LTM locations, grouped by retrocue neutral and matching trials. **d** Means and distributions of perceptual accuracy benefits for WM and LTM locations matching retrocues. Error bars in **b**–**d** represent ±1 *SEM* across participants (*n* = 44). In **b**, **c**, *p* values are reported according to two-way ANOVAs or post-hoc *t* tests with Bonferroni corrections. Shape stimuli in **a** are adapted from Li, A. Y., Liang, J. C., Lee, A. C. H., & Barense, M. D. (2020). The validated circular shape space: Quantifying the visual similarity of shape. *Journal of Experimental Psychology*: General, 149(5), 949–966. https://doi.org/10.1037/xge0000693.

($F$(1, 118) = 113.287, $p$ < 0.001, partial $\eta^2$ = 0.490). Response times did not show statistically significant differences between memory condition ($F$(1, 29) = 0.271, $p$ = 0.603) but the two factors interacted ($F$(1, 118) = 12.643, $p$ < 0.001, partial $\eta^2$ = 0.097). Post-hoc *t* tests showed that RT benefits were present for both WM ($t$(118) = 10.080, $p_{Bonferroni}$ < 0.001, *d* = 0.318) and LTM ($t$(118) = 9.381, $p_{Bonferroni}$ < 0.001, *d* = 0.253) but were stronger for WM ($t$(118) = 3.556, $p$ < 0.001, *d* = 0.267). For retrieval accuracy, retrocueing ($F$(1, 118) = 55.519, $p$ < 0.001, partial $\eta^2$ = 0.320) and memory condition ($F$(1, 118) = 6.947, $p$ = 0.010, partial $\eta^2$ = 0.056) both exerted significant main effects and interacted ($F$(1, 118) = 15.216, $p$ < 0.001, partial $\eta^2$ = 0.114). Post-hoc comparisons revealed that informative retrocues significantly reduced shape reproduction errors for both WM items ($t$(118) = 9.581, $p_{Bonferroni}$ < 0.001, *d* = 0.540) and LTM items ($t$(118) = 2.792,

$p_{Bonferroni}$ = 0.018, *d* = 0.143). Error reduction was stronger for WM items ($t$(118) = 3.901, $p$ < 0.001, *d* = 0.474). The results from experiment 3 thus showed that attentional orienting can improve LTM retrieval accuracy when overall LTM performance is below ceiling.

## Sensory spill-over effects of prioritising WM and LTM items

The current study also yielded insights on how prioritising items in WM or LTM impacts sensory information processing at the item location.

In experiment 1 (Fig. 2c), selective prioritisation of WM and LTM memoranda by retrocues impacted perceptual discrimination of items occurring at the matching location. Perceptual discrimination accuracy showed a main effect of retrocue matching ($F$(1, 29) = 31.541, $p$ < 0.001, partial $\eta^2$ = 0.521), with superior accuracy for discriminating

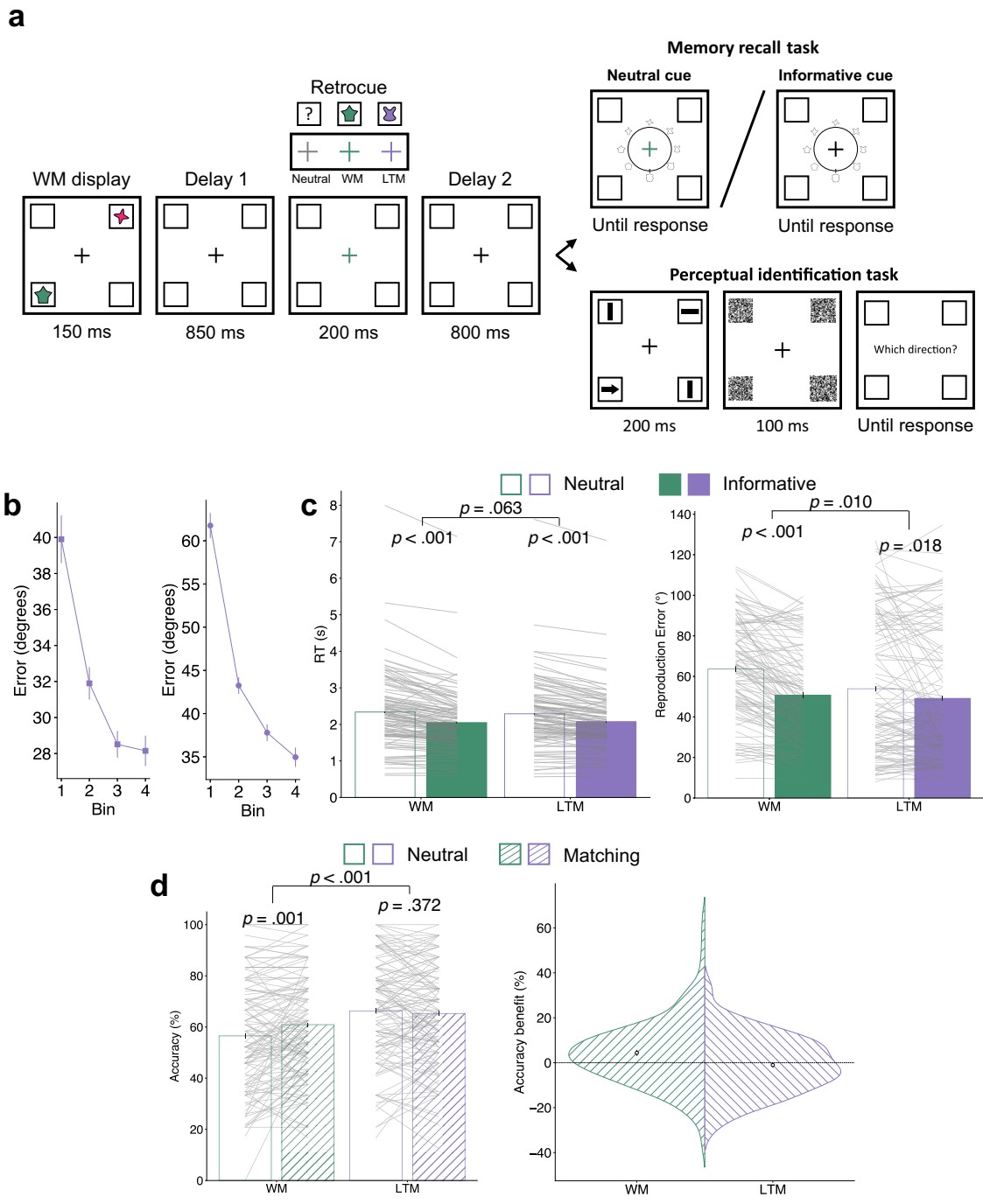

**Fig. 5 | Behavioural results in experiment 3. a** Task schematic of the testing trials in experiment 3, where only the perceptual task was different from experiment 2. In this new task, participants identified the direction of the only arrow appearing at one of the four quadrants. **b** LTM learning curves during the first day of training. Colour reproduction (left panel) and shape reproduction (right panel) errors were sorted into 4 bins (with 10 trials each). **c** Memory retrieval mean RTs and accuracy for WM and LTM items, grouped by neutral-retrocue and informative-retrocue trials. **d** Perceptual identification accuracy at WM and LTM locations, grouped by retrocue neutral and matching trials (left panel); means and distributions of perceptual accuracy benefits for WM and LTM locations matching retrocues (right panel). Error bars in **b**–**d** represent ±1 *SEM* across participants (*n* = 119). In **c**, **d**, *p* values are reported according to two-way ANOVAs or post-hoc *t* tests with Bonferroni corrections. Shape stimuli in **a** are adapted from Li, A. Y., Liang, J. C., Lee, A. C. H., & Barense, M. D. (2020). The validated circular shape space: Quantifying the visual similarity of shape. *Journal of Experimental Psychology: General*, 149(5), 949–966. https://doi.org/10.1037/xge0000693.

stimuli at retrocue-matching locations than for stimuli on neutral-retrocue trials. This accuracy benefit was significantly larger than zero for both memory conditions (WM: 0.094 ± 0.020, $t(29) = 4.804$, $p < 0.001$, $d = 0.877$; LTM: 0.121 ± 0.028, $t(29) = 4.362$, $p < 0.001$, $d = 0.797$) and there was no statistically significant difference in size ($t(29) = 0.962$, $p = 0.344$). We also observed a main effect of memory condition ($F(1, 29) = 4.887$, $p = 0.035$, partial $\eta^2 = 0.144$). Accuracy was

higher when LTM locations were probed (0.655 ± 0.026) compared to WM locations (0.601 ± 0.032). A separate analysis found no statistically significant effect of congruence between item location and arrow-pointing direction ($t(29) = 0.894$, $p = 0.379$).

Improvement of perceptual discrimination in experiment 1 could have resulted from response preparation related to reporting item location rather than prioritisation of an item in WM or LTM. In

experiment 2 (Fig. 4c), there was no direct link between spatial location and responses in the memory retrieval task, so that any effect on perceptual discrimination could be linked to prioritisation of items in WM or LTM. Under these conditions, perceptual discrimination still benefitted at locations matching the retrocued items ($F(1, 43) = 9.504$, $p = 0.004$, partial $\eta^2 = 0.181$). Perceptual benefits for items matching retrocued WM and LTM locations were both significantly larger than zero (WM: $0.037 \pm 0.003$, $t(43) = 2.191$, $p = 0.034$, $d = 0.330$; LTM: $0.032 \pm 0.002$, $t(43) = 2.176$, $p = 0.035$, $d = 0.328$) and there was no statistically significant difference in size ($t(43) = 0.259$, $p = 0.797$). As in experiment 1, discrimination performance was better at LTM than WM locations overall ($F(1, 43) = 55.213$, $p < 0.001$, partial $\eta^2 = 0.562$). A separate analysis found no statistically significant effect of congruence between item location and arrow-pointing direction ($t(43) = 0.354$, p = 0.725).

In experiment 3 (Fig. 5d), the main effect of retrocue matching did not reach significance ($F(1, 118) = 3.473$, $p = 0.065$). Memory condition still exerted a main effect, with perceptual discrimination better at LTM than WM locations ($F(1, 118) = 32.213$, $p < 0.001$, partial $\eta^2 = 0.214$). Retrocueing and memory condition interacted ($F(1, 118) = 10.328$, $p = 0.002$, partial $\eta^2 = 0.080$). Perceptual benefits of WM retrocues were significantly larger than zero ($t(118) = 3.308$, $p = 0.001$, $d = 0.214$), while benefits from LTM retrocues were not ($t(118) = 0.896$, $p = 0.372$).

### Attentional orienting in WM but not LTM elicited significant directional gaze biases

In experiment 2, eye tracking investigated the engagement of the oculomotor system during attentional orienting. Replicating previous findings[14–16], WM retrocues elicited significant gaze biases in the direction of attentional orienting. In contrast, no statistically significant evidence of similar gaze biases occurred when orienting attention in LTM (Fig. 6a–d). Compared to the baseline pattern of gaze shifts following neutral cues, systematic changes were observed for trials with WM retrocues depending on the direction of the retrocued item, both horizontally and vertically. No statistically significant changes occurred for trials with LTM retrocues, on which the baseline pattern of gaze shifts was observed regardless of the retrocued item direction.

The differences between gaze biases following WM and LTM retrocues were quantified using the associated towardness time courses. Cluster-based permutation testing identified significant differences following WM and LTM retrocues in both horizontal and vertical directions (horizontal: ~450–1000 ms after cue onset, $p = 0.002$; vertical: ~420–870 ms after cue onset, $p = 0.003$). No statistically significant clusters occurred following LTM retrocues.

A complementary analysis of directional microsaccades (see ref. 42) showed a similar pattern (Fig. 6e, f). After WM retrocues, microsaccades occurred more often in the direction of the prioritised item than the other item (~380–490 ms after cue onset, $p < 0.001$). No statistically significant directional changes in microsaccade rates occurred after LTM retrocues.

## Discussion

We demonstrate that internal attention-directing cues benefit both WM and LTM recall in humans, with qualitatively different consequences, and through non-coextensive mechanisms. Our findings reinforce the increasing recognition of the strong and multidirectional relationship between attention and memories of different durations[43,44] as well as the plurality of mechanisms supporting their interactions[7].

We introduced an approach for investigating and comparing selective attention operating in LTM vs. WM using equivalent stimulus parameters and response requirements within the same tasks. In our tasks, orienting attention to contents in LTM significantly improved retrieval speed in a similar, though less pronounced way, as for WM.

Accuracy benefits were more consistent for WM than LTM. No statistically significant improvements in LTM accuracy occurred when retrieval performance was at very high levels overall (experiments 1 and 2), but improvements emerged when there was greater room for improvement in retrieval accuracy (experiment 3). The current pattern of results suggests that attention can, at least, consistently enhance the accessibility or "readiness" to act on LTM memory representations. The modulations observed for WM compared to LTM retrieval were stronger, shortening response times to a greater degree and consistently affecting retrieval accuracy. The findings suggest differences in the mechanisms of internal attention within these different memory domains. However, on their own, these behavioural results could result from variations in the strength of the memory representations themselves and the resulting retrieval demands.

Our eye-tracking data yielded the most suggestive evidence for a functional dissociation between the mechanisms for orienting attention in WM and LTM. By measuring directional biases in eye gaze and microsaccades, we revealed a significant difference in oculomotor involvement in trials with WM vs. LTM retrocues. Consistent with previous work[14–16], we observed that gaze was biased toward retrocued items in WM. However, no statistically significant gaze biases were detected following LTM retrocues.

Our eye-tracking results thus clearly demonstrate that internal attention in LTM is not co-extensive with internal attention in WM. Orienting attention in WM engages a control network of frontal, parietal and subcortical areas modulating activity in task-relevant sensory and motor areas[45–47]. The close relationship between the internal attention control regions and regions involved in oculomotor control results in correlated gaze shifts and microsaccades. The network and mechanisms for orienting attention in LTM have not yet been fully characterised. They may be qualitatively different than the network for attention in WM. For example, posterior parietal areas have been implicated in selective LTM retrieval, with parallels drawn to external attention mechanisms (e.g., ref. 29). However, the degree of anatomical overlap has been questioned[33]. Engagement of attention-related frontal areas has been less conspicuous. In addition, established plasticity patterns related to the longer-term associations may confer alternative or additional mechanisms for prioritising feature values of LTM items. Informative LTM retrocues may interact with latent functional states that have been intrinsically reinforced by associative plasticity.

We also measured the spill-over effects of orienting in WM and LTM on sensory processing. While previous reports of this phenomenon focused on the influence of WM[48,49], our results showed that comparable effects can result from prioritising LTM representations. In the perceptual discrimination task, prioritising items within the spatial layout of either WM or LTM boosted the discrimination accuracy of location-matching stimuli in two of the three experiments (experiments 1 and 2). This occurred despite there being no strategic benefit for using the retrocue information in the perceptual task, since the location of the retrocued item only matched the probed location on 25% of the trials. Consequences of memory prioritisation for sensory processing were therefore likely incidental. These perceptual improvements at the matching locations of attended items reflect the engagement of spatial attention. It is worth noting that spatial representations are not strictly required in the Testing phase of experiment 2; cueing the item colour required the retrieval of the corresponding shape. The automatic recruitment of spatial attention, even when not required, has been commonly observed in WM tasks[14,50]. The spatial layout of encoded items has been proposed to provide an important scaffolding for WM representations[51–53]. The findings suggest that the spatial layout may also be a useful or intrinsic preserved property of encoded LTM arrays.

Interestingly, in experiment 2, significant perceptual benefits at LTM-matching locations were observed despite no significant

## Gaze position: horizontal direction

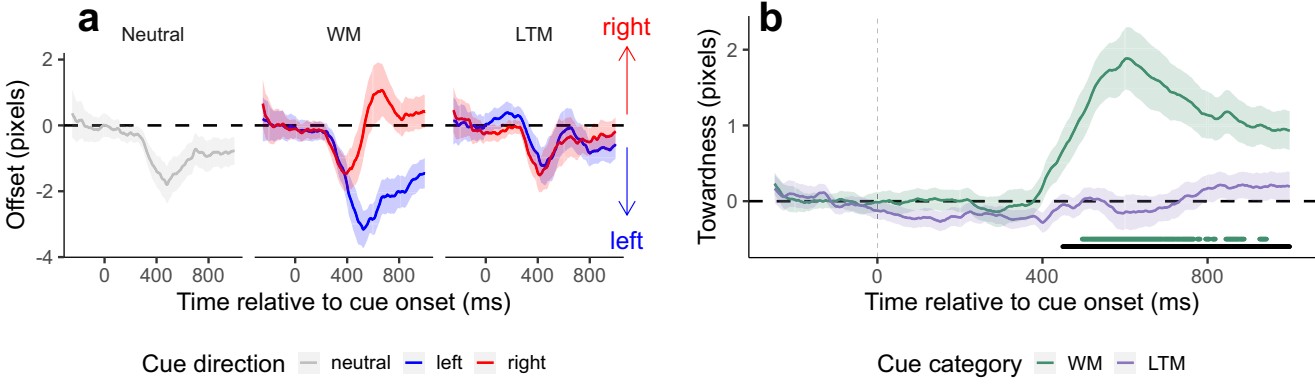

## Gaze position: vertical direction

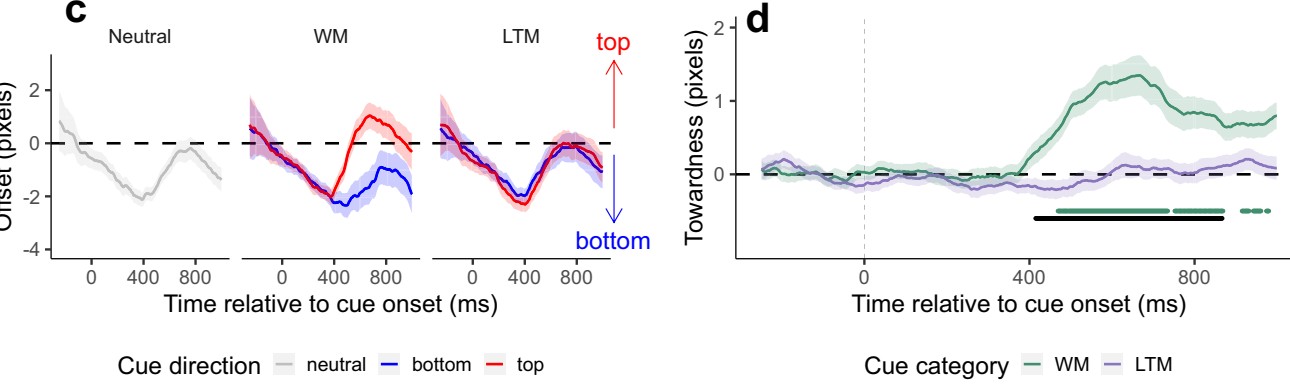

## Saccade rate: horizontal direction

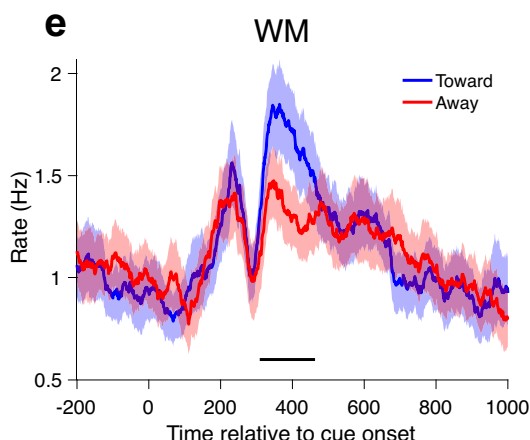
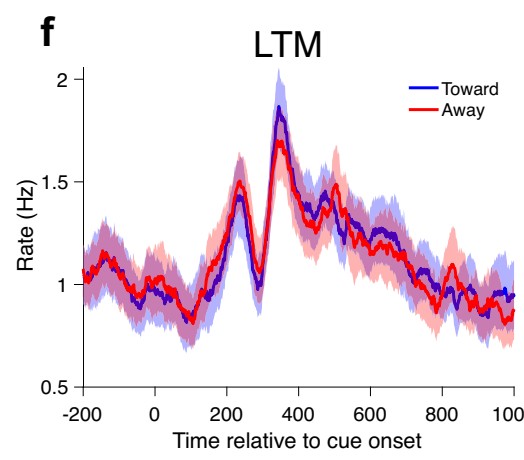

**Fig. 6 | Eye-tracking results in experiment 2. a** Time courses of horizontal gaze biases following neutral, WM and LTM retrocues. For WM and LTM retrocues, time courses are shown separately for conditions in which the retrocued item occupied the left side (top/bottom left location) or the right side (top/bottom right location) during encoding. **b** Time courses of horizontal gaze bias towardness on trials with WM and LTM retrocues, corresponding to the data in **a**. **c** Time courses of vertical gaze biases following neutral, WM and LTM retrocues. For WM and LTM retrocues, time courses were shown separately for conditions in which the retrocued item occupied the top side (top left/right location) or the bottom side (bottom left/right

location) during encoding. **d** Time courses of vertical gaze bias towardness on trials with WM and LTM retrocues, corresponding to the data in **c**. **e** Time courses of toward vs. away microsaccade rate in the horizontal channel, following WM retrocues. **f** Time courses of toward vs. away microsaccade rate in the horizontal channel, following LTM retrocues. Horizontal lines below the time courses in **b**, **d** and **e** indicate significant temporal clusters (green lines: WM compared to zero; black lines in **b** and **d**: WM vs. LTM difference; black line in **e**: toward vs. away difference). Shading areas in the time courses in **a**–**f** represent ±1 SEM.

differences in gaze biases or microsaccades. That is, focusing within LTM facilitated sensory discrimination in a spatially specific way without accompanying spatially specific oculomotor readouts. We consider multiple possible explanations for this finding. First, previous

studies have shown that oculomotor markers are strongly correlated with orienting attention within the spatial layout of WM but are not obligatory[42]. The current findings might indicate that the correlation between internal spatial attention and eye movements is weaker when

attention is directed to LTM contents. Furthermore, the nature of the contents stored may undergo transformations over the memory life-span, shifting from more visual codes reflecting sensory attributes to more abstract codes including verbal or semantic dimensions (e.g., refs. 54–57). A combination of these accounts is also possible, suggesting that focusing within LTM may rely on more flexibly activating the most relevant attributes of prior experiences to guide behaviour, whereas WM representations are necessarily spatially tethered.

Experiment 3 introduced a simpler perceptual identification task to corroborate further the findings on perceptual benefits from the previous two experiments. Retrocueing WM items still conferred perceptual benefits, whereas retrocueing LTM items no longer did. The lack of perceptual benefits linked to prioritising items in LTM could reflect the weaker nature of the next-day LTM representations in experiment 3 compared to experiments 1 and 2. Alternatively, spillover effects may have been overridden by capture from the single arrow cue or dampened by reduced competition in the perceptual discrimination task used in experiment 3 (see ref. 58). Greater variability of online performance may also have contributed. It will be interesting to explore further the boundary conditions for perceptual spillover effects of internal attention in LTM.

An additional, unexpected, and interesting finding was the superior performance in the perceptual discrimination at locations associated with LTM compared to WM items across all three experiments. Involvement of longer-term plasticity mechanisms for the LTM items may have contributed to better sensory performance for items appearing in LTM locations. Cortical plasticity mechanisms akin to operating in perceptual[59] or statistical[60] learning could render the functional states associated with LTM locations and features more accessible to re-activation. Another possible explanation for the disparity in perceptual sensitivity at the locations of LTM vs. WM items is the degree of active sensory recruitment supporting the two types of memory (see ref. 45). Stronger activation of feature-specific sensory signals during attention to WM could interfere with processing other incoming visual stimuli. However, this explanation would not account for better performance for discriminating stimuli at attended compared to unattended WM locations. The mechanisms behind the intriguing findings warrant further study.

Some of the findings in the current study inevitably depend on specific task parameters and demands. The LTM representations investigated in this study were simple visual associations of object features and locations (e.g., ref. 61,62). These representations are similar to those often studied in visual WM tasks but are less often considered in LTM studies. Given the unimodal and simple nature of these associations, they may not require the involvement of multisensory or contextual association hubs, such as the medial temporal lobe[63]. Using these simple visual associative LTMs, we observed that selective feature retrieval did not engage oculomotor markers of spatial attention. However, under some conditions, focusing within LTM may rely on spatial codes. For example, systematic eye-movement changes accompany attention to remembered items within scenes or contexts[26,64,65]. Therefore, whereas retrieval in visual WM tasks often invokes spatial codes even when they are not strictly necessary for task performance, the dependence on spatial frameworks for retrieving information from LTM may depend on the task requirements. Our specific task conditions yielded a clear difference in the reliance on spatial codes when retrieving contents from WM vs LTM. However, under other circumstances, where spatial retrieval is an integral part of the LTM task, the dissociation may not have been apparent. For example, spatial codes may be used in both WM and LTM when spatial retrocues are used or when participants are required to retrieve or reproduce the locations of stimuli within their original contexts in WM or LTM.

Nevertheless, some of the observations in the study carry generalisable lessons. The dissociation between the degree of reliance on spatial codes when orienting attention in WM vs. LTM in our task shows that focusing on contents within LTM is not necessarily mediated by placing representations within WM (as suggested by many studies[36,39,45,66,67]) and using standard internal attention mechanisms for WM. Further research is warranted to explore whether this dissociation extends to non-spatial or more complex forms of LTM. It will also be interesting and important to build on our laboratory-based study to test for the generalisability of our observations within natural, immersive contexts, where psychological mechanisms can sometimes deviate from those predicted by experiments in more controlled, artificial settings (see ref. 68).

The nature of the spatial frameworks used when orienting attention in WM is open to further investigation. In contrast to the extensive literature considering how various representational frameworks contribute to human performance based on LTM (e.g., refs. 69–71), the topic is underexplored in WM tasks (with notable exceptions, e.g., refs. 72–74). Recent virtual-reality tasks have shown that orienting attention to items within WM in immersive settings engages both an egocentric spatiotopic framework and an allocentric framework based on the layout of the initially encoded items[16]. Further investigation will likely reveal a much richer array of interacting spatial frameworks in WM tasks.

In summary, we compared the effects of focusing attention within human WM vs. LTM and discovered that both bring significant benefits to retrieval and subsequent sensory processing, but through dissociable mechanisms. The distinct oculomotor signatures of covert spatial attention in WM and LTM corroborate a plurality of functional properties when memories of different durations guide adaptive behaviour and open novel opportunities for furthering our understanding of the relationship between WM and LTM.

## Methods
### Experiment 1
**Participants.** Thirty individuals (23 females, 7 males, $M = 25.13$ years, $SD = 4.15$) with reported normal or corrected visual acuity volunteered and received monetary compensation for participation. The experiment was approved by the Central University Research Ethics Committee of the University of Oxford, and all participants provided informed consent before any experimental procedure began.

**Apparatus and stimuli.** Participants sat in front of a 27″ monitor (1920 × 1080 pixels, 100 Hz) and rested their chin on a chinrest placed 95 cm away from the monitor. The experiment was programmed in MATLAB (MathWorks, Natick, MA, USA) using the Psychophysics Toolbox[75]. Stimuli appeared overlaid on a grey background. Throughout the experiment, four squares (2.5° in diameter) were always presented as placeholders at the four quadrants, at 5° horizontally and vertically from the central fixation to the centre of each square. The stimuli consisted of four equiluminant colours (brown [183.5, 113, 19], green [65, 143, 110.5], lilac [138, 117.5, 190], magenta [245.5, 37, 112.5]) drawn from a circle in CIELAB colour space.

**Procedure and design.** The experiment included a learning session and a testing session, separated by a 5-minute break. During the learning session, participants were trained to encode two colours and their corresponding locations into LTM (Fig. 1). These two colours were randomly selected from the four colours defined above, and they were always located along one of the two pairs of diagonal locations (i.e., top left and bottom right, or top right and bottom left, counterbalanced across participants). We refer to this pair of locations as LTM locations. Each learning trial began with a fixation display lasting randomly between 800 and 1000 ms, after which the LTM display was presented for 150 ms. Following a delay of 850 ms, participants were probed to reproduce either the colour at one location or the location of one colour.

On colour reproduction trials, a colour wheel (containing 360 colours) was presented at the centre, and participants responded by rotating the dial and selecting a colour along the wheel. The colour wheel was presented in a random orientation on every trial. Immediately after the response, feedback was presented for 1000 ms in the form of an integer ranging from 0 to 100, with 100 indicating a perfect reproduction of the probed colour and 0 indicating the exact opposite on the wheel. On location reproduction trials, one of the two colours was presented at the centre and participants responded by pressing one of four keys mapped to the four locations (Q for top left, W for top right, A for bottom left and S for bottom right). Performance feedback was presented for 500 ms, indicating whether the chosen location was correct or incorrect. Each to-be-learned attribute (two colours and two locations) was probed on 20 trials, resulting in a total of 80 learning trials presented in random order.

During the testing session, participants performed either a memory retrieval task or a perceptual discrimination task on each trial (Fig. 2a). Each testing trial began with a fixation display (800–1000 ms). A WM display followed, where the two colours unused in the learning session were presented for 150 ms at the unused pair of diagonal locations. We refer to this pair of locations as WM locations. To make sure the contents in WM were not fixed across trials, each WM colour was randomly assigned to one of the WM locations on every trial. Following a delay of 850 ms, a retrocue that was either neutral or informative was presented for 200 ms. The retrocue was neutral on one-third of the trials; the fixation display changed to white, providing no information about the item to be probed. The retrocue was informative on two-thirds of the trials; the fixation display changed to one of the four colours, matching either a WM or LTM item with equal probability. Informative retrocues indicated the item to be probed in the memory retrieval task with 100% validity. Following a second delay of 800 ms after the retrocue, the memory retrieval task and the perceptual discrimination tasks were equally likely to be presented.

In the memory retrieval task, participants were required to retrieve the location of the probed item. On trials containing an informative retrocue, participants reported the location of the WM or LTM item indicated by the retrocue. On trials containing a neutral cue, the probed item was indicated by a centrally presented colour, chosen randomly from the WM or LTM colours. Responses were delivered by pressing one of four keys mapped to the four locations (the same keys as used in the location reproduction task during the learning session). Feedback was then presented for 500 ms, indicating whether the response was correct or wrong.

In the perceptual discrimination task, four arrows (length: 1.25°, width of the tail: 0.625°, RGB value: [128, 128, 128]) were presented in the placeholders for 100 ms, after which randomly generated Gaussian noise masks were applied to the four locations for 100 ms. Following the mask, one location was probed, and participants pressed one of the arrow keys to report the arrow direction at that location. Each of the four locations had an equal possibility of being probed. The arrow direction at each location was independently drawn from four possible directions (↑, ←, ↓, →), resulting in $4^4 = 256$ combinations. The choice of presenting arrow stimuli at all four locations and using a post-cue to elicit a response was intended to avoid the sensory capture by stimuli with different attributes.

Participants were informed that the memorised items and the retrocue bore no predictive relation concerning the location of the perceptual item to be discriminated. When retrocues were informative, however, participants always needed to maintain the retrocued item in mind for potential future use because of the randomisation of the memory retrieval and perceptual discrimination trial order, which allowed us to examine any spill-over benefits elicited by the retrocue on the perceptual discrimination task. The relationship between the perceptual discrimination task and the retrocue was totally incidental because all locations were equally likely to be probed, no matter which location the retrocue would preferentially bias attention to. As a result, only 25% of informative-retrocue trials in the perceptual discrimination task were "matching" trials on which the probed sensory location coincided with the location of the retrocued item.

The testing session consisted of 480 trials divided into 10 blocks (each including 48 trials). To become familiarised with the task, participants performed an additional 48 practice trials before testing.

**Behavioural analysis.** Behavioural data were analysed in MATLAB. For the learning session, we examined the average colour reproduction errors and location reproduction accuracy, respectively, by sorting the learning trials of each type into 4 bins (each containing 10 trials). Colour reproduction errors (in units of degrees) were calculated by taking the absolute difference between the angle of the target colour and the reproduced colour on the colour wheel. One-way repeated-measures ANOVAs with linear contrast weights ([−3, −1, 1, 3]) across four bins tested for the efficacy of training. For the testing session, data from the memory retrieval and perceptual discrimination tasks were analysed separately. During pre-processing, we excluded trials on which RTs were 3 $SD$ above the individual mean across all conditions in either task. After this exclusion step, an average of 98.35% ($SD = 0.44\%$) trials were retained in the analyses.

To test for benefits of internal selective attention on WM and LTM retrieval, we analysed the average RTs and accuracy for the memory retrieval task as a function of retrocueing (neutral vs. informative) and the memory timescale of probed items (WM vs. LTM). To examine whether orienting attention to a memory item benefited subsequent perceptual processing at matching locations, we compared perceptual discrimination accuracy on retrocue matching trials vs. neutral trials (matching vs. neutral) when locations associated with WM or LTM items were probed (WM vs. LTM). To gauge the quality of perception, accuracy was the dependent variable of interest, but RTs were also evaluated for completeness.

When comparing behavioural performance between conditions, we applied a repeated-measures ANOVA and reported partial $\eta^2$ as a measure of effect size. For post hoc $t$ tests, we reported Bonferroni-corrected $p$ values that we denoted as "$p_{Bonferroni}$". We reported Cohen's $d$ as a measure of effect size for all the $t$ tests. Where relevant, the within-subject standard error of the mean (SEM) was calculated from normalised data[41]. When evaluating potential perceptual benefits elicited by WM and LTM retrocues, we applied one-sample $t$ tests against 0. All $t$ tests were two-tailed.

## Experiment 2
**Participants.** A total of 44 volunteers (27 females, 17 males, $M = 25.93$ years, $SD = 4.51$) with reported normal or corrected-to-normal visual acuity were recruited. The experiment was approved by the Central University Research Ethics Committee of the University of Oxford, and all participants provided informed consent before any experimental procedure began. The sample size was calculated using G*Power[76] to achieve 90% power for the one-sample $t$ tests performed to test the significance of perceptual benefits following WM and LTM retrocues. The effect sizes for these comparisons in experiment 1 were 0.877 and 0.797. We assumed a conservative approach and aimed to power for the detection of a medium effect size (0.5) because we expected that the manipulation in experiment 2 would lead to a smaller effect due to the incidental nature of the spatial attributes in the task. Most of the experimental setup was identical to experiment 1, with the following modifications.

**Apparatus and stimuli.** Eye movements were recorded with the Eye-Link 1000 Desktop Mount (SR Research, Ottawa, ON, Canada) at 1000 Hz. Eye-tracker calibration used the built-in calibration and validation protocols from the EyeLink software. When possible, horizontal

and vertical gaze positions were continuously recorded for both eyes. For some participants ($N = 13$), only one eye was tracked due to a lack of good-quality binocular tracking (mostly because of wearing glasses). Stimuli appeared on a white background. Four shapes were equidistantly sampled from the validated circular shape (VCS) space[77], and then randomly assigned to each WM and LTM item, adding a new feature dimension to the existing configurations in experiment 1. These same four shapes were used across all participants, but the shapes assigned to WM and LTM items were randomised across participants.

**Procedure and design.** During the learning session, participants were trained to memorise the colours and shapes of the two LTM items (Fig. 3). On every trial, they were probed to reproduce either the colour or the shape of one item. At the response stage, either a colour wheel or a shape wheel was presented at the centre, indicating the feature dimension to be reproduced in this trial. The spatial location of the item was used to probe the colour or shape reproduction, but participants were never asked to report the item location. Both the colour and shape wheels were presented in a random orientation each time. The shape wheel consisted of 360 shapes from the VCS space. To avoid clustering, eight shapes sampled from equidistant positions on the wheel were displayed along the cardinal axes (i.e., every 45 degrees). These eight shapes served as visual anchors, which were also randomly chosen every time based on the orientation of the shape wheel. Participants responded using a computer mouse that controlled the dial on the wheel. Participants had unlimited time to retrieve the item from memory and to decide what to reproduce. However, once they started moving the dial, they had only 2500 ms to complete their reproduction. This was intended to encourage participants to retrieve the exact colour or shape before moving the dial. The position of the dial when participants clicked the left mouse button or when the time limit was reached was taken as the response. Immediately after their response, participants received feedback for 500 ms. Each colour and shape was probed on 20 trials, resulting in a total of 80 learning trials presented in random order.

During the testing session, each WM shape was randomly combined with one of the WM colours on every trial (Fig. 4a). As in experiment 1, participants performed either a memory retrieval task or a perceptual discrimination task on each trial equiprobably. In the memory retrieval task, participants reproduced the shape of the item matching the retrocued colour or a randomly probed colour when the retrocue was neutral. The shape wheel was identical to that used during the learning session and was randomly rotated across trials. Uninformative, grey neutral retrocues appeared on one-fifth of the trials. The remaining trials contained informative coloured retrocues matching each of the LTM or WM colours with equal probability. In contrast to experiment 1, the memory retrieval task in experiment 2 was non-spatial. Retrieving the item shape based on the cued colour does not require using any spatial association. Although spatial locations were used for training the colour-shape associations, participants were never cued or asked to report the item location in the memory retrieval task.

The perceptual discrimination task was the same as that in experiment 1.

The testing session consisted of 600 trials divided into 10 blocks (each including 60 trials). To become familiarised with the task, participants performed an additional 30 practice trials before testing.

**Behavioural analysis.** The analyses of interest were basically the same as in experiment 1, with location reproduction in the learning and testing sessions replaced by shape reproduction. Shape reproduction errors (in units of degrees) were calculated by taking the absolute difference between the angle of the target shape and the reproduced

shape on the shape wheel. The RTs in the memory retrieval task were calculated as the time from probe onset to when the response was recorded, either when participants clicked the left mouse button or when the time limit was reached. After excluding memory retrieval and perceptual discrimination trials on which RTs were 3 $SD$ above the individual mean across all conditions, an average of 98.66% ($SD = 0.47\%$) trials were retained in the analyses.

**Eye-tracking analysis.** Data were first converted from edf to asc format and subsequently read into RStudio. For binocularly tracked participants, data from the left and right eyes were averaged to obtain a single horizontal and a single vertical gaze position channel. Blinks were marked by detecting NaN clusters in the eye-tracking data and then interpolated using a linear interpolation procedure. Data were epoched from 250 ms before to 1000 ms after cue onset. To make our analyses more robust to the drift of the eyes during fixation, we obtained the average gaze position within a 250-ms window before cue onset for every trial and participant and subtracted it from every corresponding time course. We performed analyses on horizontal and vertical channels separately. For both channels, we only included trials on which gaze position remained within ±50% from fixation (with 100% denoting the centres of the original item locations at a ±5° visual angle) throughout the trial, as previous work showed that the gaze bias phenomenon is constituted by a bias in gaze around fixation[14–16]. For the horizontal channel, data for all 44 participants were used, with an average of 9.6% ± 1.7% ($M \pm SEM$) trials excluded per participant. For the vertical channel, six participants had to be removed due to a high number of excluded trials (>50%). For the 38 participants retained, an average of 14.6% ± 2.1% trials were excluded. Gaze time courses were smoothed using a 25-ms average moving window.

For the horizontal channel, we compared trial-averaged gaze-position time courses between conditions in which the retrocued item occupied the left side (top/bottom left location) or the right side (top/bottom right location) during encoding, separately for trials with WM retrocues and trials with LTM retrocues. For the vertical channel, we did the same between conditions in which the retrocued item occupied the bottom side (bottom left/right location) or the top side (top left/right location) during encoding, also separately for trials with WM and LTM retrocues. For both channels, trial-averaged gaze-position time courses were also obtained for trials with neutral retrocues. To increase sensitivity, we also constructed a measure of towardness separately for trials with WM and LTM retrocues, which expressed the gaze bias toward the side of the retrocued item in a single value[14–16]. We did this for both horizontal and vertical channels to obtain towardness in horizontal and vertical directions, respectively.

Analysis of microsaccade rates toward vs. away from prioritised items in WM or LTM followed previously described methods[42]. Saccades were detected using a velocity-based method, and only those with a magnitude smaller than 1 degree of visual angle were considered microsaccades[42]. Furthermore, for direct comparisons with previous work[42,78], we focused exclusively on microsaccades along the horizontal axis. Depending on the side of the retrocued item, we labelled microsaccades as "toward" or "away" based on whether they were moving toward or away from the retrocued item. The resulting time courses of "toward" and "away" microsaccade rates were smoothed using a moving average with a 50-ms sliding window.

Statistical evaluation of the towardness time courses used a cluster-based permutation approach[79] implemented in the *permuco* package, which is ideally suited to evaluate physiological effects across multiple time points while retaining high sensitivity.

## Experiment 3
**Participants.** Given the assumption that LTM would decay on the second day, we aimed to power for detecting an effect size of 0.3 for perceptual benefits, which led G*power to yield a sample size of 119

participants. Participants (86 males, 33 females, $M = 28.69$ years, $SD = 6.04$; 106 right-handed, 12 left-handed, 1 ambidextrous) were recruited on Prolific (https://www.prolific.co/). They were pre-screened on demographic criteria (age range 18 to 40, fluent in English), general health (normal or corrected-to-normal vision, no previous or ongoing mental illnesses), and participation history on Prolific Academic (participated in at least 10 studies, with a study approval rate above 90%). The experiment was approved by the Central University Research Ethics Committee of the University of Oxford. All participants provided informed consent before participating and were paid $12 per hour.

**Stimuli and procedure.** Participants performed 80 learning trials on the first day, which was identical to the learning session in experiment 2. Participants were invited to complete the testing session around 24 hours after training. The differences in the testing session compared to experiment 2 are as follows.

Experiment 3 used a simpler perceptual discrimination task, which avoided post-cues. The motivation was to tap into perceptual sensitivity more directly, without requiring post-encoding transfer and selection before behavioural reports. In the perceptual array, only one placeholder contained an arrow drawn from four directions ($\uparrow, \leftarrow, \downarrow, \rightarrow$). The other three placeholders contained randomly selected horizontal or vertical bars as control stimuli (Fig. 5a). Correspondingly, at the probe stage, participants were asked to report the direction of the single presented arrow by pressing an arrow key. Considering the greater variability of stimulus timing in online studies[80], the duration of the perceptual array was lengthened to 200 ms. The testing session consisted of 480 trials divided into 12 blocks (each including 40 trials). Participants performed an additional block of 40 practice trials before the 480 testing trials. During pre-processing, we excluded memory retrieval and perceptual discrimination trials with RTs exceeding 3 $SD$ above the individual mean across all conditions. After this procedure, an average of 98.65% ($SD = 0.61\%$) trials were retained in the analyses.

**Reporting summary**
Further information on research design is available in the Nature Portfolio Reporting Summary linked to this article.

## Data availability
The raw behavioural and eye-tracking data are publicly available through the Open Science Framework at https://doi.org/10.17605/osf.io/n629s[81].

## Code availability
Analysis code is available at https://github.com/Daniel-Gong/Orienting-in-WM-and-LTM/. A Zenodo version is also available at https://doi.org/10.5281/zenodo.14968822[82].

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

## Acknowledgements

This research was funded by a Clarendon Scholarship, a Medical Research Council Studentship and a New College-Yeotown Scholarship to D.G.; a Wellcome Trust Senior Investigator Award (104571/Z/14/Z) and a James S. McDonnell Foundation Understanding Human Cognition Collaborative Award (220020448) to A.C.N.; and by the NIHR Oxford Health Biomedical Research Centre. The Wellcome Centre for Integrative Neuroimaging is supported by core funding from the Wellcome Trust (203139/Z/16/Z). For the purpose of open access, the author has applied a CC BY public copyright licence to any Author Accepted Manuscript version arising from this submission.

## Author contributions

D.G., D.D. and A.C.N. conceived and designed the experiments. D.G. acquired and analysed the data. D.G., D.D. and A.C.N. interpreted the data. D.G., D.D., and A.C.N. draughted and revised the manuscript.

## Competing interests

The authors declare no competing interests.
