## [Transparent Peer Review file · Nature Communications]

Focusing attention in working and long-term memory through dissociable mechanisms

Corresponding Author: Mr Dongyu Gong

Version 0:

Reviewer comments:

Reviewer #1

(Remarks to the Author)

This study explores the role of spatial coding in working memory and long-term memory using a clever retro-cue design that allows the authors to examine memory performance in both tasks, and measures whether spatial orienting – if present – spills over to an unrelated perceptual discrimination task. I enjoyed reading the paper: The question is interesting, the measures well designed, and the experiment and analyses executed well. However, several questions and concerns arose which I am outlining below.

My first concern is basic: The main question of interest is whether LTM representations also use a spatial code to store and retrieve features, as has been shown in WM. To me it seemed that the task set up here really pushes people to store LTM items in a spatial way though: this is especially true in Exp 1 that asks people to recall spatial locations (and the authors acknowledge this and add Exp 2); but even in Exp 2, the LTM training task uses spatial retrieval cues, really asking participants to link the items to spatial positions on the display, in my view, in a relatively explicit way. My question is simply whether this set up allows us to draw any conclusions about LTM representations in general, or only how they are learned and probed in this particular design? To me this also raised many questions about whether these spatial effects in these memory systems are likely retinotopic or spatiotopic.... –there is lots of research on these questions that seems relevant but that was not discussed at all.

Relatedly, I am not sure if the results of the training task are the result of consolidation into long-term memory. All trials are repeated exactly the same way, same color/shape at the same location. So couldn't the performance increase also be due to simple priming, or encoding items multiple times which strengthens working memory representations? It this would also consistent with better performance. How do we know this task creates "true" long-term memories? It seems that maybe a longer delay, ideally paired with an interference task that eliminates any possibility of active maintenance of these colors and locations would be critical. The working memory task inserted during 'test' does not directly interfere with these representations as different locations are used. One way to show that these items are stored in LTM, not WM, could be to assess the representations using manipulations known to affect long-term memories differently than working memories, for example proactive interference. (e.g., If these memories are robust to proactive interference, that would bolster the argument that these memories are truly consolidated in long-term memory.)

Exp 1 has several issues, including the explicit spatial recall (which makes the results not particularly surprising or interesting), but also the differences in difficulty induced by the distinct memory probes between spatial locations (4AFC) vs. color (360AFC). Many of these concerns are alleviated by Exp 2 but while reading the paper, I felt skeptical of several aspects of Exp 1. Thus, I would recommend to inform the reader earlier on about an Exp 2 and build up an anticipation for it.

While Exp 2 is clearly much stronger than Exp 1, the training task itself still uses spatial locations to explicitly probe memories at the end of the trial in a spatial manner. I am not sure it's fair to say that spatial locations were completely irrelevant and encoded entirely incidental.

In Exp 2, color reproduction errors do not improve significantly and remain overall relatively high (~18degr.) relative to Exp 1 (~10degr.). Why is this? Is there a trade-off in remembering color and shape vs. color and location? (More overlap in resources)? Or is it simply that the surface area of the shapes is smaller on average and so participants simply don't encode the color as well overall because of less activation in the respective neural populations? I am not sure whether the current

data allows to tease these hypotheses apart but it seems curious and should at a minimum be discussed.

The differences in gaze bias between WM and LTM is interesting, in particular given the perceptual discrimination task data. It would be nice to hear more about what processes each measure taps into and why the authors think this dissociation arose. Furthermore, I was interested in seeing whether the gaze biases in WM are mostly driven by microsaccades or not? I know in earlier papers this research group has often reported both; it seems important to do the same here, as this may inform hypotheses about the functionality of these gaze biases during WM recall.

I fully agree with the authors that it seems critical to investigate whether the null effects observed in some conditions are due to overall ceiling performance or a 'true' difference between WM and LTM. However, I was less satisfied by just leaving this up to 'future studies', given that this was the main question the paper started with; at least for this outlet, I felt like these concerns need to be directly addressed. I realize asking to run another experiment to match performance across tasks can be a lot; however, as it is, it seems unclear whether the observed differences are due to different memory systems or differences in task difficulty (this is also reinforced by my earlier points where I am questioning whether the training really creates long-term memories per se).

A final question about the perceptual discrimination task: It seemed like an interesting choice to use arrows as stimuli; I am wondering whether you can separate out 'incongruent' and 'congruent' trials (e.g., arrow on the left pointing to the right, arrow at the top pointing to the bottom vs. arrow on the left pointing to the left, arrow at the top pointing to the top, etc.). It seems like behavioral performance could be strongly influenced by that and it seems like if that's the case, you'd really want to make sure these are fully counterbalanced across various conditions.

The abstract states that "Attentional orienting improved memory retrieval for both memory types" – that doesn't seem quite right? In both experiments, the accuracy benefit was only present for the WM task, not the LTM task?

Minor:

Page 8: Is this actually supplement or part of the main text? It wasn't clear to me.
Was the response time data cleaned at all before computing the average RT?

Reviewer #2

(Remarks to the Author)

Dong and colleagues present an interesting study attempting to contrast effects of selective attention in working memory (WM) with those in long-term memory (LTM) in a couple of retro-cuing experiments where cued stimuli are either pre-learned (LTM) or recently encoded (WM). Data from subsequent memory probes and a perceptual task (probing memory biases) suggest highly similar consequences of attentional selection in WM and LTM, with a notable difference being gaze biased toward cued location at retro-cue onset, which are only present for the WM condition. This study addresses an important topic using a creative task design. However, in my opinion, some shortcomings of said design render the conclusions for most of the key behavioral comparisons inconclusive, as they do not provide a clean contrast between LTM and WM retrieval. I am not certain that these issues can be resolved without collecting additional data.

1- The introduction generally provides a nice and coherent set-up for the rest of the paper, but one key aspect makes the conceptual advance that could be carved out here rather blurry. In particular, the stipulation that selective attention can "modulate" LTM retrieval does not seem to have a plausible null-hypothesis. How else would we be able to retrieve specific LTMs at will? I do not know of anyone in the field of memory research who does not assume that retrieval from LTM can be selective and intentional, and is under those circumstances driven by internally directed attention. It doesn't even matter whether people call this attention or not (some may prefer "memory retrieval") – the mere fact that we are clearly capable of selective, goal-directed retrieval of LTMs means that the answer to the question "is attention involved in LTM retrieval?" is a foregone conclusion. Therefore, I don't think that a "demonstration that attention can be flexibly directed to specific LTM items" (quoted from the discussion) is particularly meaningful, since we already know that. The degree to which that selection shares characteristics with the attentional selection of items in WM is a more interesting question, and that is what the present study seems to be addressing. Asking "does attention do something" vs. "is attention's role equivalent in the two cases" are two different questions though, and it would be beneficial to more clearly disentangle them in the introduction (and beyond).

2- My major concern with the authors' interpretation of the study results is that I don't think the behavior during the memory and perceptual probes can discriminate between LTM and WM retrieval very cleanly, particularly for the retro-cued trial. The problem is that there is an 800ms delay between the retro-cue and the probes. I would argue (and I think many people in the WM field would agree) that when the retro-cue cues an LTM item, the most likely scenario is that this leads to said item being retrieved from LTM to WM (because now the subject knows she'll need to use that item to guide behavior, a classic role ascribed to WM, including by some of the current authors). Thus, by the time the memory or perceptual probes come around, even in the retro-cued LTM condition, the cued items are in fact most likely held in WM. Only for neutral trials (no retro-cues) does the probe onset require instant attentional selection (retrieval) from LTM when LTM items are probed. Under this view, only the neutral cue condition behavior can reliably distinguish between attentional selection from WM vs. LTM. In support of this assumption, the mean performance data, and reported condition differences (especially the interaction effects) in fact

seem to be driven primarily (or at least more substantially) by differences in the neutral cue conditions rather than in the retro-cued case (see e.g., Figs. 2B and 3B). This perspective is also supported by the eye gaze results of Experiment 2: the eye gaze measurements are time-locked to the retro-cue, such that they allow for a clean comparison between attentional selection in WM and LTM retrieval. In this condition there are clear differences detected between the WM and LTM conditions, whereas such differences are absent or much less pronounced in the behavioral probes (800ms later) – again suggesting that by that time, the cued LTM items have likely been retrieved into WM. (It would be interesting to examine eye gaze effects at probe, as here the two conditions might look more equivalent for retro-cued trials). In my view, the authors need to refute this alternative interpretation in order for their conclusions to stand up, but I can't see a way of doing that without conducting additional experiments.

3- Experiment 2 has a nice design feature that controls for the possibility that specific responses are being maintained in WM following the retro-cue. This seems worth highlighting in the introduction to Experiment 2 (rather than just noting it in the general discussion).

Reviewer #3

(Remarks to the Author)

In this manuscript, authors reported two experiments comparing the memory performance benefits resulting from the internal prioritization of relevant items between working memory (WM) and long-term memory (LTM). In addition, they tested whether the internal shift of attention impacts the processing of sensory input that is irrelevant to a memory task. The experiments revealed the presence of retro-cue benefits reflected in reaction times (RT) for both WM and LTM. However, RT benefits were stronger for WM than LTM. Moreover, accuracy benefits occurred only for WM but not for LTM. The results also showed that shifting attention to the relevant memory item improved the discriminating accuracy of visual stimuli at matching locations. Lastly, eye-tracking data showed that gaze biases related to orienting attention are elicited following retro-cues in WM but not in LTM.

Experiment 2 is well-designed (Experiment 1 has a confound), the analytical approach is sound, and the results are novel and interesting. There are some weaknesses—the paper largely avoided discussion of mechanism and did not have much to say about the LTM literature.

The authors should consider whether Experiment 1 is worth including. In general, showing a replication is beneficial, but Experiment 1 has a design flaw. Heightened perceptual sensitivity could be due to the response being location based (thankfully it's a button press and not a mouse-click, but pressing a button that has been mapped to a location might enhance perceptual sensitivity at that location). Participants can plan a response upon seeing the cue, such that it's impossible to know if the enhancement is response or selection-based. Further, there did not seem to be any conclusions that rested entirely on Experiment 1. If included in a revision, the authors should discuss the above concern in the transition between experiments 1 and 2, rather than subtly in the discussion.

Cutting Experiment 1 could leave more page space to devote to something the manuscript is lacking—discussion of what the results might tell us about the underlying mechanism or its implications for LTM theory. There is a rich literature on how we access representations in LTM. The manuscript would be stronger if it considered what implications the present work had for that literature. Further, the discussion of the results mostly just restated the findings. Here are some examples where I wanted deeper insight. On p.21 the authors wrote, "The findings could point to differences in the types of mechanisms of internal attention within these different memory domains...". What potential types of mechanisms do authors have in mind? On p.22 authors wrote, "results may be tapping into something more fundamental, such as relevant LTM representations exerting stronger and more automatic biases, akin to sensory salience effects". Are the authors suggesting that LTM in the current design is akin to priming (or statistical learning), and that is automatic and requires minimal effort?

Related to this, what type of LTM representation do the authors believe the task is encouraging? It is worth noting that, in our daily lives, the location of long-term memory representations is rarely as critical as for this task (for this reason, the location effects of LTM are really quite surprising). Therefore, it's worth a comment on the real-world validity of this task and discussion of what the implications are for other aspects of LTM such as recognition. Further, I suspect the same training results could have been found even without ever showing the stimulus, given that this was trained with feedback.

Another important question is whether the perceptual sensitivity effects are linked causally with selection, or whether these are epiphenomenal effects or demand characteristics. To look at this, the authors could test if there is a correlation between the amount of retrocue benefits and the change in perceptual sensitivity (e.g. participants with the largest retrocue effects also show the most changes in perceptual sensitivity).

Why did the authors use a postcue during the perceptual discrimination task? This means that effects could be less about changes in perception per se and more about transference of perceptual input into a stable form that can be reported. At the least, this decision should be justified in the revised manuscript.

Minor points:

1. The abstract could describe the results in more detail to provide information about differences in retro-cue benefits between WM and LTM.
2. For potential replication, some information should be added to the method section of Experiment 1 (describing the learning phase): what was used to respond to a color wheel? What was the duration of the feedback? Which keys were used

to respond to the location?

3. On p. 7/8, I believe the word "supplementary information" was not intended to be formatted this way. Further, the callout to the supplemental could be more specific, summarizing what was found. I agree with keeping the details in the supplemental but a sentence or two about non-matching items would be appropriate. For example, one might wonder whether LTM representations could benefit both items, since they are always shown together.

4. In both figures, it is difficult to see retro-cue colors. Maybe the fixation could be scaled larger. Further, it should be emphasized more that the fixation colors are referring to specific LTM or WM items. Otherwise, the figure did a good job of conveying the method details.

5. Have the authors considered that a framework like sensory recruitment could predict worse perceptual sensitivity for WM relative to LTM locations (since resources at that location would be devoted to the memory item).

6. Using the phrase memory timescales when discussing ANOVA results etc. was confusing to read. Perhaps memory condition is better.

Version 1:

Reviewer comments:

Reviewer #1

(Remarks to the Author)

This revision addressed most of my concerns well.

I appreciate that the authors collected more data and clarified several points in the manuscript, in particular those pertaining to what kind of long-term memories (LTM) this study taps into, and the issue regarding the ecological validity of the task (which was raised by several reviewers).

Overall, I remain a bit skeptical that the current set of experiments supports the rather general claim that WM and LTM processes rely on separate mechanisms. This might certainly be true in the current task, but re-reading the paper (together with the reviews) did not alleviate my concerns that this is a highly specific effect that pertains to this particular (and somewhat artificial) paradigm where only spatial strategies were tested and measured. It's long been known that in visual working memory spatial codes are particularly robust and important to maintain and retrieve information, which has not been true for LTM, where spatial codes are simply not that relevant (agreeing with Rev #3 here). The response to previous reviews confirmed that the authors also think that this study tests a very specific kind of LTM, namely "simple visual associations of object features and locations" that do not require the involvement of the "medial temporal lobe". This left me wondering whether the authors can conclude that LTM retrieval is different from WM in a more general sense that would advance theories of LTM, or whether these findings test a niche case of LTM after all. It certainly appears that the current data only allow conclusions that pertain to the SPATIAL coding between WM and LTM, and this is a much more narrow conclusion than currently put forward, and also a much more nuanced theoretical advancement.

(Remarks on code availability)

Reviewer #2

(Remarks to the Author)

The authors implemented quite extensive revisions, including an additional experiment, and these efforts have strengthened the paper in terms of narrowing down the precise impacts of, and differences between, cuing of attention in LTM vs. WM. I think the set of findings coming out of these experiments is novel and noteworthy, and will find an interested readership (though an experimental psychology journal would be a more obvious outlet in this regard). One minor remaining shortcoming is that the discussion largely re-describes the results rather than pursuing a bit more theorizing. For instance, the main finding seems to be that LTM representations of the colored shape stimuli are either disconnected from their spatial location or from the oculomotor system (or both). These observations are reported in the discussion section but not discussed very deeply in terms of why and how this might occur. For instance, perhaps the spatial information in LTM has been semanticized/translated into a verbal code? Other possibilities exist, and at least this reader would have liked to hear more informed speculation about the implications of these findings.

(Remarks on code availability)

Reviewer #4

(Remarks to the Author)

I have carefully reviewed the revised manuscript and the authors' responses and have concluded that they have adequately addressed all the concerns. Although I remain skeptical about including Exp1, I understand that the authors prefer to retain it. Importantly, the shortcomings of Exp1 are clearly described.

The new Experiment 3 is a valuable addition to the study. The elimination of the post-cue, as the authors noted, more directly taps into perceptual sensitivity, making the interpretations more straightforward. Moreover, the long delay between the training and testing sessions in Experiment 3 allows for ruling out priming as an alternative explanation for the observed results. This significantly strengthens the conclusions in the manuscript.

Lastly, the authors provide a detailed discussion of the possible mechanisms of internal attention in LTM, further enhancing the manuscript.

At this point, I have no further concerns and believe the manuscript is ready for publication in its current form.

(Remarks on code availability)

REVIEWER COMMENTS

Reviewer #1 (Remarks to the Author):

This study explores the role of spatial coding in working memory and long-term memory using a clever retro-cue design that allows the authors to examine memory performance in both tasks, and measures whether spatial orienting – if present – spills over to an unrelated perceptual discrimination task. I enjoyed reading the paper: The question is interesting, the measures well designed, and the experiment and analyses executed well. However, several questions and concerns arose which I am outlining below.

Thank you for your positive evaluation of our manuscript and for the important points you raised.

My first concern is basic: The main question of interest is whether LTM representations also use a spatial code to store and retrieve features, as has been shown in WM. To me it seemed that the task set up here really pushes people to store LTM items in a spatial way though: this is especially true in Exp 1 that asks people to recall spatial locations (and the authors acknowledge this and add Exp 2); but even in Exp 2, the LTM training task uses spatial retrieval cues, really asking participants to link the items to spatial positions on the display, in my view, in a relatively explicit way. My question is simply whether this set up allows us to draw any conclusions about LTM representations in general, or only how they are learned and probed in this particular design? To me this also raised many questions about whether these spatial effects in these memory systems are likely retinotopic or spatiotopic... –there is lots of research on these questions that seems relevant but that was not discussed at all.

Thank you for raising this concern. It was not our main intention to probe whether LTM representations use a spatial code. However, we concede that our task designs (especially in Experiment 1) make spatial codes conspicuous and potentially relevant. We had two main aims: (1) to test whether LTM performance benefited from focused attention and (2) to compare the benefits of focused attention in LTM vs. WM. If, as many assume, attention in LTM works through bringing the representation into WM, then we would expect similar behavioral performance benefits during memory retrieval as well as consequences for sensory processing. Our findings show a non-overlapping pattern of effects, suggesting at least some independence between focusing in LTM vs. WM. Like many previous WM studies (e.g., Schneegans & Bays, 2017; Sreenivasan et al., 2014; Thom et al., 2023; van Ede et al., 2019), we observed clear spatial mechanisms for focusing attention in the WM task using eye-tracking measures even though spatial information was not strictly necessary for task performance (cues used color and reports involved shape). Strikingly, this reliance on spatial codes was not observed for the LTM conditions in the same experiment.

Our findings hold for the types of experiments we have conducted. The specific patterns of attention benefits may vary according to the nature of LTM representation/association and task demands. However, the demonstration of a functional dissociation between attention-related benefits in WM and LTM is a core generalizable finding with important consequences. We have noted the limited nature of conclusions we can draw based on this set of findings in the discussion:

“Some of the findings in the current study inevitably depend on specific task parameters and demands. The LTM representations investigated in this study were simple visual associations of object features and locations (e.g., Balaban et al., 2020; Shimi & Logie, 2019). These representations are similar to those often studied in visual working-memory tasks but are less often considered in LTM studies. Given the unimodal and simple nature of these associations, they may not require the involvement of multisensory or contextual association hubs, such as the medial temporal lobe (see Sanders & Cowell, 2023). Using these simple visual associative LTMs, we observed that selective feature retrieval did not engage oculomotor markers of spatial attention. However, under some conditions, focusing within LTM may rely on spatial codes. For example, systematic eye-movement changes accompany attention to remembered items within scenes or contexts (e.g., Hayhoe et al., 1998; Henderson & Hollingworth, 2003; Summerfield et al., 2006).

Nevertheless, some of the observations in the study carry generalizable lessons. The dissociation between the degree of reliance on spatial codes when orienting attention in WM vs. LTM in our task shows that focusing on contents within LTM is not necessarily mediated by placing representations within WM and using standard internal attention mechanisms for WM...” (pages 12-13)

The question regarding the type of spatial frames that support focused attention in WM is interesting but not well explored to date. Using virtual reality, we have recently demonstrated that different types of spatial representations can be involved (Draschkow et al., 2022). We have now discussed this:

“The nature of the spatial frameworks used when orienting attention in working memory is open to further investigation. In contrast to the extensive literature considering how various representational frameworks contribute to human performance based on LTM (e.g., Committeri et al., 2004; Kerkhoff, 2001; Moraresku & Vlcek, 2020), the topic is underexplored in WM tasks (with notable exceptions, e.g., Aagten-Murphy & Bays, 2019; Golomb & Kanwisher, 2012; Shafer-Skelton & Golomb, 2018). Recent virtual-reality tasks have shown that orienting attention to items within working memory in immersive settings engages both an egocentric spatiotopic framework and an allocentric framework based on the layout of the initially encoded items (Draschkow et al, 2022). Further investigation will likely reveal a much richer array of interacting spatial frameworks in WM tasks”. (page 13)

Relatedly, I am not sure if the results of the training task are the result of consolidation into long-term memory. All trials are repeated exactly the same way, same color/shape at the same location. So couldn't the performance increase also be due to simple priming, or encoding items multiple times which strengthens working memory representations? It this would also consistent with better performance. How do we know this task creates “true” long-term memories? It seems that maybe a longer delay, ideally paired with an interference task that eliminates any possibility of active maintenance of these colors and locations would be critical. The working memory task inserted during ‘test’ does not directly interfere with these representations as different locations are used. One way to show that these items are stored

in LTM, not WM, could be to assess the representations using manipulations known to affect long-term memories differently than working memories, for example proactive interference. (e.g., If these memories are robust to proactive interference, that would bolster the argument that these memories are truly consolidated in long-term memory.)

We took this comment to heart and conducted a new experiment. Experiment 3 separated the training and retro-cueing tasks across days, thereby ruling out access to other, shorter-term types of memory in the LTM condition. Experiment 3 provided a clear comparison of the consequences of focusing attention in LTM vs. WM.

To clarify our procedures in the original experiments (1 & 2): Our training task was intended to build reliable LTM representations. It is possible, and indeed likely, that priming-related effects contributed to the learning. The retro-cueing task (in both Experiments 1 and 2) followed the training task after an interval of approximately five minutes, during which the experimenter conversed with and distracted the participant. It is important to note that the LTM items are never presented again during the retro-cueing (testing) phase of the experiments. Performance in trials in which LTM representations are probed is based on the previous training session. Performance in LTM-probed trials during the retro-cueing task remained high despite the items not being presented again, providing good evidence that the trained associations from the training phase are retained in LTM.

To eliminate the possibility that shorter-term priming effects were responsible for our results, we conducted a new experiment online. In this version of the experiment, participants completed the training on one day and completed the retro-cueing (testing) task on the subsequent day. Experiment 3 replicated the main pattern of benefits by selective attention in LTM vs. WM. The quality of LTM performance on day 2 remained high, again showing that the training task did a good job building high-quality LTMs.

Exp 1 has several issues, including the explicit spatial recall (which makes the results not particularly surprising or interesting), but also the differences in difficulty induced by the distinct memory probes between spatial locations (4AFC) vs. color (360AFC). Many of these concerns are alleviated by Exp 2 but while reading the paper, I felt skeptical of several aspects of Exp 1. Thus, I would recommend to inform the reader earlier on about an Exp 2 and build up an anticipation for it.

Thank you for your suggestion. We have now more clearly informed the reader about what to expect from the experimental designs before going into details of individual experiments. The first paragraph in the results presents a synthesis of how the experiments complement one another.

“Across three experiments, we demonstrate benefits conferred by orienting attention in WM and LTM, which differed qualitatively. In Experiment 1, participants acquired two color-location bindings in LTM during a training session. In a subsequent testing session, they encoded two additional color-location bindings into WM on every trial. Half of the trials tested for benefits of orienting attention in WM or LTM. Participants indicated the location of an item based on informative or non-informative color retrocues. The remaining half tested the impact of prioritizing an item location in WM or LTM on

sensory processing. Participants discriminated the direction of one of four arrows briefly flashed at WM and LTM locations. Experiments 2 and 3 followed the same training-testing approach, with testing trials measuring benefits of internal attention on memory performance or sensory processing. Experiment 2 extended the findings to color-shape bindings. A non-spatial task probed for benefits of internal attention. Participants reproduced item shapes using a randomly oriented wheel after informative or non-informative color retrocues. This shape reproduction method controlled for the possible contributions of motor preparation in memory retrieval performance. The perceptual discrimination trials were the same as in Experiment 1. Experiment 3 was based on Experiment 2, with training and testing sessions separated across two successive days to ensure that LTM effects did not reflect lingering priming or shorter-term memory traces. The perceptual discrimination task was replaced with a simpler task. Only one arrow appeared, and participants identified its direction.” (page 5)

While Exp 2 is clearly much stronger than Exp 1, the training task itself still uses spatial locations to explicitly probe memories at the end of the trial in a spatial manner. I am not sure it's fair to say that spatial locations were completely irrelevant and encoded entirely incidentally.

This is a fair comment. Our intention was to indicate that retrieving spatial location is not strictly required for the retro-cueing task in Experiment 2, in which we probe the consequences of focusing attention in WM vs. LTM. The reviewer is correct to note that the training task requires the use of spatial location and may have led to the deliberate encoding of spatial associations. We have made this point clearer when describing Experiment 2 in the methods.

“... The spatial location of the item was used to probe the color or shape reproduction, but participants were never asked to report the item location...” (page 18)

“... In contrast to Experiment 1, the memory retrieval task in Experiment 2 was non-spatial. Retrieving the item shape based on the cued color does not require using any spatial association. Although spatial locations were used for training the color-shape associations, participants were never cued about or asked to report the item location in the memory-retrieval task.” (pages 18-19)

In Exp 2, color reproduction errors do not improve significantly and remain overall relatively high (~18degr.) relative to Exp 1 (~10degr.). Why is this? Is there a trade-off in remembering color and shape vs. color and location? (More overlap in resources)? Or is it simply that the surface area of the shapes is smaller on average and so participants simply don't encode the color as well overall because of less activation in the respective neural populations? I am not sure whether the current data allows to tease these hypotheses apart but it seems curious and should at a minimum be discussed.

Thanks for bringing up this question. We do not have a firm explanation for why performance for reproducing colors was better in the training of Experiment 1. The reason why color reproduction is better in Experiment 1 may simply reflect the simpler nature of the stimuli (varying only in the color dimension), but we cannot conclude that for sure. The new Experiment 3 uses the same training methods as Experiment 2 and yields improvements for

both shape and color reports. However, as expected for an online experiment, learning performance is less accurate overall. In the description of the results, we now state:

“In Experiment 2 (Fig. 3), shape reproduction improved significantly across bins (linear contrast: $F(1, 40) = 17.939, p < .001, \text{partial } \eta^2 = 0.122$). Although improvement in color reproduction was not significant ($F(1, 40) = 2.304, p = .132$), the errors remained relatively low with a numerically decreasing trend across bins (first bin: $23.107^\circ \pm 2.295$, last bin: $19.182^\circ \pm 1.612$). In Experiment 3 (Fig. 6b), linear contrasts showed that learning performance improved significantly for both color reproduction ($F(1, 115) = 9.015, p < .001, \text{partial } \eta^2 = 0.054$) and shape reproduction ($F(1, 115) = 38.470, p < .001, \text{partial } \eta^2 = 0.196$).” (pages 5-6)

The differences in gaze bias between WM and LTM is interesting, in particular given the perceptual discrimination task data. It would be nice to hear more about what processes each measure taps into and why the authors think this dissociation arose. Furthermore, I was interested in seeing whether the gaze biases in WM are mostly driven by microsaccades or not? I know in earlier papers this research group has often reported both; it seems important to do the same here, as this may inform hypotheses about the functionality of these gaze biases during WM recall.

Thank you for this comment and for nudging us to include the microsaccadic data. Indeed, the strong differences in oculomotor behavior were the most revealing data showing that attention in LTM vs. WM does not always rely on overlapping mechanisms.

We have now supplemented the gaze-position analysis with an additional analysis of microsaccades. The microsaccade findings showed a similar pattern to the gaze analysis. When WM items were cued, we observed a greater proportion of microsaccades toward the memorized location of the cued item (approximately 380-490 ms after cue onset, $p < .001$). In contrast, there was no difference in toward vs. away microsaccades when LTM items were cued.

The directional gaze and microsaccade biases accompanying attention in WM replicate previous studies reporting oculomotor measures of internal attention in WM (de Vries et al.,

2023; Linde-Domingo & Spitzer, 2024; Liu et al., 2022, 2023; van Ede et al., 2019). In contrast, when LTM items were cued, the proportion of microsaccades toward vs. away from the cued items was balanced. Previous investigations of the relationship between microsaccades and oscillatory markers of internal attention have suggested that microsaccades and alpha modulation are strongly correlated but not obligatorily linked (Liu et al., 2022). Microsaccades and gaze shifts may be a readout of spatial attention orienting due to the close relationship between the oculomotor and covert attention systems (see Nobre et al., 1997, 2000). However, engagement of oculomotor plans may not be necessary for orienting attention (Liu et al., 2022).

Our speculative interpretation regarding the differential engagement of oculomotor processes is that spatial location provided a more useful scaffolding for selectively focusing on WM representations than LTM representations. In LTM representations, the link between color and shape in Experiment 2 appears to have been sufficient to support performance.

As the reviewer notes, it is interesting to consider the differential pattern of oculomotor behavior in light of the subsequent impact on the perceptual discrimination task. Despite the presence vs. absence of systematic gaze biases, both types of internal attention conferred similar benefits to sensory processing. Such a finding supports the view that there is a non-obligatory relationship between microsaccades and attention. We did not elaborate on this important point, but have now addressed this directly in the revised manuscript:

“Interestingly, in Experiment 2, significant perceptual benefits at LTM-matching locations were observed despite no significant differences in gaze biases or microsaccades. Previous studies have shown that oculomotor markers are strongly correlated with orienting attention within the spatial layout of WM but are not obligatory (Liu et al., 2022). The current findings reinforce the non-obligatory nature of the relationship between internal spatial attention and eye movements. Focusing within LTM can facilitate sensory discrimination in a spatially specific way without accompanying oculomotor readouts.” (page 11)

I fully agree with the authors that it seems critical to investigate whether the null effects observed in some conditions are due to overall ceiling performance or a ‘true’ difference between WM and LTM. However, I was less satisfied by just leaving this up to ‘future studies’, given that this was the main question the paper started with; at least for this outlet, I felt like these concerns need to be directly addressed. I realize asking to run another experiment to match performance across tasks can be a lot; however, as it is, it seems unclear whether the observed differences are due to different memory systems or differences in task difficulty (this is also reinforced by my earlier points where I am questioning whether the training really creates long-term memories per se).

Thank you for this comment. The new study helps us address whether the null effects are due to overall ceiling performance. In Experiment 3, training took place on Day 1 and testing was on the next day. Under these conditions, performance in the LTM condition was still good but below ceiling. Performance in the LTM reproduction task in Experiment 3 was worse than in Experiment 2 (reproduction errors in Experiment 3 vs. Experiment 2: 51.547 ± 2.880 vs. 17.378 ± 1.819 , Welch’s $t = 10.031$, $p < .001$). Unlike in Experiment 2, LTM retrocues significantly reduced reproduction errors relative to the neutral-retrocue condition, suggesting that the null

effect in Experiment 2 is likely due to the ceiling-level performance in the neutral-retrocue condition.

A final question about the perceptual discrimination task: It seemed like an interesting choice to use arrows as stimuli; I am wondering whether you can separate out ‘incongruent’ and ‘congruent’ trials (e.g., arrow on the left pointing to the right, arrow at the top pointing to the bottom vs. arrow on the left pointing to the left, arrow at the top pointing to the top, etc.). It seems like behavioral performance could be strongly influenced by that and it seems like if that’s the case, you’d really want to make sure these are fully counterbalanced across various conditions.

This was a point worth checking. We had assumed that our balancing of the conditions would account for any possible Simon effect, but we had not checked that this was indeed the case. We now compared performance on the perceptual discrimination task when the direction of the arrow was congruent vs. incongruent with the location of the cued item. There was no significant effect of congruency in Experiments 1 and 2 (Experiment 1: $t(29) = 0.894, p = 0.379$; Experiment 2: $t(43) = 0.354, p = 0.725$). We noted the lack of congruency effects in the current results.

The abstract states that “Attentional orienting improved memory retrieval for both memory types” – that doesn’t seem quite right? In both experiments, the accuracy benefit was only present for the WM task, not the LTM task?

Consistent improvements in the speed of memory retrieval were observed for LTM items, but we agree we could have been more careful and precise with our wording. Taking into account the results from the three experiments, we have reworded the statement in the abstract:

“Attentional orienting benefited performance for both WM and LTM, with stronger effects for WM.” (Abstract)

Page 8: Is this actually supplement or part of the main text? It wasn’t clear to me. Was the response time data cleaned at all before computing the average RT?

Thank you for spotting this. The text “Supplementary Information” appeared at the end of a sentence in the main text, but the formatting made it a bit confusing. We have now corrected the formatting.

Yes, response-time data were pre-processed to exclude outliers. This is described in the Methods section:

Experiment 1: “During pre-processing, we excluded trials on which RTs were 3 SD above the individual mean across all conditions in either task. After this exclusion step, an average of 98.35% (SD = 0.44%) trials were retained in the analyses.” (page 16)

Experiment 2: “After excluding memory retrieval and perceptual discrimination trials on which RTs were 3 SD above the individual mean across all conditions, an average of 98.66% (SD = 0.47%) trials were retained in the analyses.” (page 19)

Experiment 3: “During pre-processing, we excluded memory retrieval and perceptual discrimination trials with RTs exceeding 3 SD above the individual mean across all conditions. After this procedure, an average of 98.65% (SD = 0.61%) trials were retained in the analyses.” (page 21)

Reviewer #2 (Remarks to the Author):

Gong and colleagues present an interesting study attempting to contrast effects of selective attention in working memory (WM) with those in long-term memory (LTM) in a couple of retro-cuing experiments where cued stimuli are either pre-learned (LTM) or recently encoded (WM). Data from subsequent memory probes and a perceptual task (probing memory biases) suggest highly similar consequences of attentional selection in WM and LTM, with a notable difference being gaze biased toward cued location at retro-cue onset, which are only present for the WM condition. This study addresses an important topic using a creative task design. However, in my opinion, some shortcomings of said design render the conclusions for most of the key

behavioral comparisons inconclusive, as they do not provide a clean contrast between LTM and WM retrieval. I am not certain that these issues can be resolved without collecting additional data.

We thank the reviewer for the important points that have helped us focus and strengthen the core contributions of the study.

1- The introduction generally provides a nice and coherent set-up for the rest of the paper, but one key aspect makes the conceptual advance that could be carved out here rather blurry. In particular, the stipulation that selective attention can “modulate” LTM retrieval does not seem to have a plausible null-hypothesis. How else would we be able to retrieve specific LTMs at will? I do not know of anyone in the field of memory research who does not assume that retrieval from LTM can be selective and intentional, and is under those circumstances driven by internally directed attention. It doesn’t even matter whether people call this attention or not (some may prefer “memory retrieval”) – the mere fact that we are clearly capable of selective, goal-directed retrieval of LTMs means that the answer to the question “is attention involved in LTM retrieval?” is a foregone conclusion. Therefore, I don’t think that a “demonstration that attention can be flexibly directed to specific LTM items” (quoted from the discussion) is particularly meaningful, since we already know that. The degree to which that selection shares characteristics with the attentional selection of items in WM is a more interesting question, and that is what the present study seems to be addressing. Asking “does attention do something” vs. “is attention’s role equivalent in the two cases” are two different questions though, and it would be beneficial to more clearly disentangle them in the introduction (and beyond).

We agree entirely with the reviewer. Indeed, our study asks whether selective attention in LTM relies on equivalent processes to selective attention in WM. We have reframed and cleaned up the introduction to avoid the offending statements. We now describe the central question more specifically.

“... However, less is understood about how selective attention prioritizes different contents within LTM. One major open question is whether the ability to focus attention on specific contents within LTM, to guide selective retrieval, relies on the same internal attention processes that operate in WM. Most standard models of memory suggest that LTM retrieval is mediated by WM (Atkinson & Shiffrin, 1968; Baddeley et al., 2021). If representations move to WM, then the same internal attention mechanisms should select relevant mnemonic content for behavior. However, in principle, focusing on LTM content to guide retrieval may also occur directly and independently.” (pages 3-4)

2- My major concern with the authors’ interpretation of the study results is that I don’t think the behavior during the memory and perceptual probes can discriminate between LTM and WM retrieval very cleanly, particularly for the retro-cued trial. The problem is that there is an 800ms delay between the retro-cue and the probes. I would argue (and I think many people in the WM field would agree) that when the retro-cue cues an LTM item, the most likely scenario is that this leads to said item being retrieved from LTM to WM (because now the subject knows she’ll need to use that item to guide behavior, a classic role ascribed to WM, including by some

of the current authors). Thus, by the time the memory or perceptual probes come around, even in the retro-cued LTM condition, the cued items are in fact most likely held in WM. Only for neutral trials (no retro-cues) does the probe onset require instant attentional selection (retrieval) from LTM when LTM items are probed. Under this view, only the neutral cue condition behavior can reliably distinguish between attentional selection from WM vs. LTM. In support of this assumption, the mean performance data, and reported condition differences (especially the interaction effects) in fact seem to be driven primarily (or at least more substantially) by differences in the neutral cue conditions rather than in the retro-cued case (see e.g., Figs. 2B and 3B). This perspective is also supported by the eye gaze results of Experiment 2: the eye gaze measurements are time-locked to the retro-cue, such that they allow for a clean comparison between attentional selection in WM and LTM retrieval. In this condition there are clear differences detected between the WM and LTM conditions, whereas such differences are absent or much less pronounced in the behavioral probes (800ms later) – again suggesting that by that time, the cued LTM items have likely been retrieved into WM. (It would be interesting to examine eye gaze effects at probe, as here the two conditions might look more equivalent for retro-cued trials). In my view, the authors need to refute this alternative interpretation in order for their conclusions to stand up, but I can't see a way of doing that without conducting additional experiments.

The reviewer's comments get to the essence of what we wished to test in the study. Most researchers share the reviewer's premise that LTM representations are brought into WM for retrieval. This is a core tenet of the modal model (Atkinson & Shiffrin, 1968) and probably has much older roots. Yet, our results suggest that this may not be the entire story. If selective retrieval in LTM were entirely mediated through a WM representation, we would expect a similar pattern of attention benefits and consequences for sensory processing. The behavioral patterns differed in telling ways. Most strikingly, the oculomotor behavior (Experiment 2) showed a striking difference in reliance on spatial attention mechanisms. If selective attention can only operate on LTM after it has been brought into WM, we would have seen gaze biases following LTM retrocues as well (though perhaps delayed by the time required to move items into WM).

As suggested, we have now analyzed the oculomotor behavior following the probe stimulus for retrocued trials. We note that the analysis of oculomotor behavior during the probe period is compromised by participants being free to move their eyes to inspect the response dial. The data are therefore much messier. Nevertheless, the results are still informative.

Even at the time of the probe, the differences persist. The oculomotor signals are much more pronounced and only significant for the WM trials. If LTM representations moved into WM, we would expect an equivalent pattern of gaze biases.

Cue direction — left — right

Cue category — WM — LTM

Time courses of gaze biases and towardness following probe onset in WM- and LTM-retrocued trials.

3- Experiment 2 has a nice design feature that controls for the possibility that specific responses are being maintained in WM following the retro-cue. This seems worth highlighting in the introduction to Experiment 2 (rather than just noting it in the general discussion).

Thank you for your suggestion. We have now emphasized this design feature in the introduction to Experiment 2.

“Participants reproduced item shapes using a randomly oriented wheel after informative or non-informative color retrocues. This shape reproduction method controlled for the possible contributions of motor preparation in memory retrieval performance.”(page 5)

Reviewer #3 (Remarks to the Author):

In this manuscript, authors reported two experiments comparing the memory performance benefits resulting from the internal prioritization of relevant items between working memory (WM) and long-term memory (LTM). In addition, they tested whether the internal shift of attention impacts the processing of sensory input that is irrelevant to a memory task. The experiments revealed the presence of retro-cue benefits reflected in reaction times (RT) for both WM and LTM. However, RT benefits were stronger for WM than LTM. Moreover, accuracy benefits occurred only for WM but not for LTM. The results also showed that shifting attention to the relevant memory item improved the discriminating accuracy of visual stimuli at matching locations. Lastly, eye-tracking data showed that gaze biases related to orienting attention are elicited following retro-cues in WM but not in LTM.

Experiment 2 is well-designed (Experiment 1 has a confound), the analytical approach is sound, and the results are novel and interesting. There are some weaknesses—the paper largely avoided discussion of mechanism and did not have much to say about the LTM literature.

We thank the reviewer for the insightful points and for pushing us to discuss the implications and mechanisms behind our findings in greater depth.

The authors should consider whether Experiment 1 is worth including. In general, showing a replication is beneficial, but Experiment 1 has a design flaw. Heightened perceptual sensitivity could be due to the response being location based (thankfully it's a button press and not a mouse-click, but pressing a button that has been mapped to a location might enhance perceptual sensitivity at that location). Participants can plan a response upon seeing the cue, such that it's impossible to know if the enhancement is response or selection-based. Further, there did not seem to be any conclusions that rested entirely on Experiment 1. If included in a revision, the authors should discuss the above concern in the transition between experiments 1 and 2, rather than subtly in the discussion.

We understand that Experiment 1 has some shortcomings but decided to keep it, though shortened, to emphasize the reliability of the effects. We followed the reviewer's suggestion to spell out the shortcomings:

“A shortcoming of Experiment 1 was the spatial nature of the memory-retrieval report. Upon presentation of an informative retrocue, participants could immediately prepare their response, thus confounding the quality and speed of item selection with response preparation. The non-spatial nature of Experiments 2 and 3, combined with using a randomly oriented response wheel, overcame this limitation.” (page 6)

Cutting Experiment 1 could leave more page space to devote to something the manuscript is

lacking—discussion of what the results might tell us about the underlying mechanism or its implications for LTM theory. There is a rich literature on how we access representations in LTM. The manuscript would be stronger if it considered what implications the present work had for that literature. Further, the discussion of the results mostly just restated the findings. Here are some examples where I wanted deeper insight. On p.21 the authors wrote, “The findings could point to differences in the types of mechanisms of internal attention within these different memory domains...”. What potential types of mechanisms do authors have in mind? On p.22 authors wrote, “results may be tapping into something more fundamental, such as relevant LTM representations exerting stronger and more automatic biases, akin to sensory salience effects”. Are the authors suggesting that LTM in the current design is akin to priming (or statistical learning), and that is automatic and requires minimal effort?

Although we recognize the rich and vast literature on LTM, we aimed to ask a very basic question, which has remained unexplored. Does the ability to focus attention on specific contents within LTM representations (e.g., to guide selective retrieval) rely on the same types of internal attention processes that operate on WM representations? In doing so, we also hoped to provide a new experimental approach to investigate the mechanisms of selective attention within LTM more broadly.

Our findings provide a simple new fundamental insight – that internal attention in LTM is not the same as internal attention in WM. Beyond that, there is little we can say for sure about how selective attention within LTM operates. Nevertheless, aspects of our findings do invite us to make contact with the broader LTM literature as well as to offer some reasonable speculation regarding mechanisms.

Regarding potential differences in mechanisms, we now add:

“Our eye-tracking results thus clearly demonstrate that internal attention in LTM is not co-extensive with internal attention in WM. Orienting attention in WM engages a control network of frontal, parietal, and subcortical areas modulating activity in task-relevant sensory and motor (D’Esposito & Postle, 2015; Nobre et al., 2004; Wallis et al., 2015). The close relationship between the internal attention control regions and regions involved in oculomotor control results in correlated gaze shifts and microsaccades. The network and mechanisms for orienting attention in LTM have not yet been fully characterized. They may be qualitatively different than the network for attention in WM. For example, posterior parietal areas have been implicated in selective LTM retrieval, with parallels drawn to external attention mechanisms (e.g., Cabeza et al., 2008). However, the degree of anatomical overlap has been questioned (Hutchinson et al., 2009). Engagement of attention-related frontal areas has been less conspicuous. In addition, established plasticity patterns related to the longer-term associations may confer alternative or additional mechanisms for prioritizing feature values of LTM items. Informative LTM retrocues may interact with latent functional states that have been intrinsically reinforced by associative plasticity.” (pages 10-11)

Regarding LTM representations exerting stronger and more automatic biases, we now add:

“... Involvement of longer-term plasticity mechanisms for the LTM items may have contributed to better sensory performance for items appearing in LTM locations. Cortical plasticity mechanisms akin to operating in perceptual (Watanabe & Sasaki, 2015) or statistical (Conway, 2020) learning could render the functional states associated with LTM locations and features more accessible to re-activation. Another possible explanation for the disparity in perceptual sensitivity at the locations of LTM vs. WM items is the degree of active sensory recruitment supporting the two types of memory (see D’Esposito & Postle, 2015). Stronger activation of feature-specific sensory signals during attention to WM could interfere with processing other incoming visual stimuli. However, this explanation would not account for better performance for discriminating stimuli at attended compared to unattended WM locations. The mechanisms behind the intriguing findings warrant further study.” (page 12)

Related to this, what type of LTM representation do the authors believe the task is encouraging? It is worth noting that, in our daily lives, the location of long-term memory representations is rarely as critical as for this task (for this reason, the location effects of LTM are really quite surprising). Therefore, it’s worth a comment on the real-world validity of this task and discussion of what the implications are for other aspects of LTM such as recognition. Further, I suspect the same training results could have been found even without ever showing the stimulus, given that this was trained with feedback.

We concede that our experimental tasks have low ecological validity. They were designed as controlled laboratory instruments to measure the consequences of internal attention in LTM and WM in a sensitive manner and in a way that could be compared to the literature on internal attention in WM.

We acknowledge this in the current manuscript:

“Some of the findings in the current study inevitably depend on specific task parameters and demands... It will be interesting and important to build on our laboratory-based study to test for the generalizability of our observations within natural, immersive contexts, where psychological mechanisms can sometimes deviate from those predicted by experiments in more controlled, artificial settings (see Draschkow et al., 2021).” (pages 12-13)

We now mention the type of LTM memory we studied and the resulting implications and limitations.

“... The LTM representations investigated in this study were simple visual associations of object features and locations (e.g., Balaban et al., 2020; Shimi & Logie, 2019). These representations are similar to those often studied in visual working-memory tasks but are less often considered in LTM studies. Given the unimodal and simple nature of these associations, they may not require the involvement of multisensory or contextual association hubs, such as the medial temporal lobe (see Sanders & Cowell, 2023). Using these simple visual associative LTMs, we observed that selective feature retrieval did not engage oculomotor markers of spatial attention. However, under some conditions, focusing within LTM may rely on spatial codes. For example, systematic eye-movement

changes accompany attention to remembered items within scenes or contexts (e.g., Hayhoe et al., 1998; Henderson & Hollingworth, 2003; Summerfield et al., 2006).” (page 12)

Another important question is whether the perceptual sensitivity effects are linked causally with selection, or whether these are epiphenomenal effects or demand characteristics. To look at this, the authors could test if there is a correlation between the amount of retrocue benefits and the change in perceptual sensitivity (e.g. participants with the largest retrocue effects also show the most changes in perceptual sensitivity).

The extent to which the perceptual sensitivity effects are specifically tied to reproduction benefits is worth interrogating further. We tested whether the normalized RT benefit for internal attention in WM and LTM correlated with normalized perceptual sensitivity scores. We found consistent positive slopes for both WM and LTM benefits by perceptual sensitivity in all three experiments, but the effects did not reach statistical significance. We speculate that both effects are mediated by internal attention but that other factors influencing response times in the different tasks may have also impacted response times. (We used RTs rather than accuracy since this was consistently improved by internal attention for both types of memories across all tasks.)

Why did the authors use a postcue during the perceptual discrimination task? This means that effects could be less about changes in perception per se and more about transference of perceptual input into a stable form that can be reported. At the least, this decision should be justified in the revised manuscript.

The reviewer makes a good point. The original motivation for the design was simply to avoid sensory capture by differences in visual salience across the sensory array. However, we can see that using the postcues makes it harder to interpret the locus of the modulation resulting in the perceptual benefit.

In the new Experiment 3, we took the opportunity to introduce a simpler perceptual identification task. Only one directional arrow appears in the sensory array, with the other three locations containing visually similar control stimuli.

We have now stated in the methods:

“... The choice of presenting arrow stimuli at all four locations and using a post-cue to elicit a response was intended to avoid the sensory capture by stimuli with different attributes.” (page 15)

“Experiment 3 used a simpler perceptual discrimination task, which avoided post-cues. The motivation was to tap into perceptual sensitivity more directly, without requiring post-encoding transfer and selection before behavioral reports...” (page 21)

“... Retrocuing WM items still conferred perceptual benefits, whereas retrocuing LTM items no longer did. The lack of perceptual benefits linked to prioritizing items in LTM could reflect the weaker nature of the next-day LTM representations in Experiment 3

compared to Experiments 1 and 2. Alternatively, spillover effects may have been overridden by capture from the single arrow cue or dampened by reduced competition in the perceptual discrimination task used in Experiment 3 (see Sawetsuttipan et al., 2023). Greater variability of online performance may also have contributed. It will be interesting to explore further the boundary conditions for perceptual spillover effects of internal attention in LTM.” (pages 11-12)

Minor points:

1. The abstract could describe the results in more detail to provide information about differences in retro-cue benefits between WM and LTM.

Reviewer 1 made a similar point. We have reworded the abstract to convey the pattern of results more precisely. We had to be short on details because of word limits.

“... Attentional orienting benefited performance for both WM and LTM, with stronger effects for WM. Eye-tracking revealed significant gaze shifts and microsaccades correlated with attention in WM but not LTM. Visual discrimination of unrelated visual stimuli was consistently improved for items matching attended WM locations. Similar effects occurred at LTM locations but less consistently...” (Abstract)

2. For potential replication, some information should be added to the method section of Experiment 1 (describing the learning phase): what was used to respond to a color wheel? What was the duration of the feedback? Which keys were used to respond to the location?

Thank you. We have provided more methodological details, including how color and location responses were made and the duration of the feedback.

“On color reproduction trials, a color wheel (containing 360 colors) was presented at the center, and participants responded by rotating the dial and selecting a color along the wheel. The color wheel was presented in a random orientation on every trial. Immediately after the response, feedback was presented for 1000 ms in the form of an integer ranging from 0 to 100, with 100 indicating a perfect reproduction of the probed color and 0 indicating the exact opposite on the wheel. On location reproduction trials, one of the two colors was presented at the center, and participants responded by pressing one of four keys mapped to the four locations (Q for top left, W for top right, A for bottom left, and S for bottom right). Performance feedback was presented for 500 ms indicating whether the chosen location was correct or wrong. Each to-be-learned attribute (two colors and two locations) was probed on 20 trials, resulting in a total of 80 learning trials presented in random order.” (page 14)

3. On p. 7/8, I believe the word "supplementary information" was not intended to be formatted this way. Further, the callout to the supplemental could be more specific, summarizing what was found. I agree with keeping the details in the supplemental but a sentence or two about

non-matching items would be appropriate. For example, one might wonder whether LTM representations could benefit both items, since they are always shown together.

Yes, this was a formatting error. We have corrected the formatting.

4. In both figures, it is difficult to see retro-cue colors. Maybe the fixation could be scaled larger. Further, it should be emphasized more that the fixation colors are referring to specific LTM or WM items. Otherwise, the figure did a good job of conveying the method details.

Thank you. We have made the fixation cross larger so that retrocue colors can be seen more clearly. We have also added an illustration to indicate that the retrocue colors signal specific WM or LTM items.

Improved task schematic for Exp. 1

Improved task schematic for Exp. 2

Stimuli adapted from Li, A. Y., Liang, J. C., Lee, A. C. H., & Barense, M. D. (2020). The validated circular shape space: Quantifying the visual similarity of shape. *Journal of Experimental Psychology: General*, 149(5), 949–966. <https://doi.org/10.1037/xge0000693>

5. Have the authors considered that a framework like sensory recruitment could predict worse

perceptual sensitivity for WM relative to LTM locations (since resources at that location would be devoted to the memory item).

We had not considered this perspective. We agree that the sensory recruitment hypothesis could help explain worse perceptual sensitivity at WM locations, though we don't think it can concomitantly account for the sensory benefits at the attended WM location. We have added this possibility when discussing the perceptual sensitivity effects:

“... Another possible explanation for the disparity in perceptual sensitivity at the locations of LTM vs. WM items is the degree of active sensory recruitment supporting the two types of memory (see D'Esposito & Postle, 2015). Stronger activation of feature-specific sensory signals during attention to WM could interfere with processing other incoming visual stimuli. However, this explanation would not account for better performance for discriminating stimuli at attended compared to unattended WM locations.” (page 12)

6. Using the phrase memory timescales when discussing ANOVA results etc. was confusing to read. Perhaps memory condition is better.

Yes, we can see that. We have rephrased “memory timescales” as “memory conditions” throughout the manuscript.

List of references mentioned in the response:

- Aagten-Murphy, D., & Bays, P. M. (2019). Independent working memory resources for egocentric and allocentric spatial information. *PLOS Computational Biology*, *15*(2), e1006563. <https://doi.org/10.1371/journal.pcbi.1006563>
- Atkinson, R. C., & Shiffrin, R. M. (1968). Human memory: A proposed system and its control processes. *Psychology of Learning and Motivation*, *2*, 89–195. [https://doi.org/10.1016/s0079-7421\(08\)60422-3](https://doi.org/10.1016/s0079-7421(08)60422-3)
- Baddeley, A. D., Hitch, G., & Allen, R. (2021). A multicomponent model of working memory. *Working Memory: State of the Science*, 10–43.
- Balaban, H., Assaf, D., Arad Meir, M., & Luria, R. (2020). Different features of real-world objects are represented in a dependent manner in long-term memory. *Journal of Experimental Psychology: General*, *149*(7), 1275.
- Cabeza, R., Ciaramelli, E., Olson, I. R., & Moscovitch, M. (2008). The parietal cortex and episodic memory: An attentional account. *Nature Reviews Neuroscience*, *9*(8), Article 8. <https://doi.org/10.1038/nrn2459>
- Committeri, G., Galati, G., Paradis, A.-L., Pizzamiglio, L., Berthoz, A., & LeBihan, D. (2004). Reference frames for spatial cognition: Different brain areas are involved in viewer-, object-, and landmark-centered judgments about object location. *Journal of Cognitive Neuroscience*, *16*(9), 1517–1535. <https://doi.org/10.1162/0898929042568550>
- Conway, C. M. (2020). How does the brain learn environmental structure? Ten core principles for understanding the neurocognitive mechanisms of statistical learning.

- Neuroscience & Biobehavioral Reviews*, 112, 279–299.
<https://doi.org/10.1016/j.neubiorev.2020.01.032>
- de Vries, E., Fejer, G., & van Ede, F. (2023). No obligatory trade-off between the use of space and time for working memory. *Communications Psychology*, 1(1), 1–10.
<https://doi.org/10.1038/s44271-023-00042-9>
- D’Esposito, M., & Postle, B. R. (2015). The Cognitive Neuroscience of Working Memory. *Annual Review of Psychology*, 66(1), 115–142. <https://doi.org/10.1146/annurev-psych-010814-015031>
- Draschkow, D., Kallmayer, M., & Nobre, A. C. (2021). When Natural Behavior Engages Working Memory. *Current Biology*, 31(4), 869–874.e5.
<https://doi.org/10.1016/j.cub.2020.11.013>
- Draschkow, D., Nobre, A. C., & van Ede, F. (2022). Multiple spatial frames for immersive working memory. *Nature Human Behaviour*, 6(4), Article 4.
<https://doi.org/10.1038/s41562-021-01245-y>
- Golomb, J. D., & Kanwisher, N. (2012). Higher Level Visual Cortex Represents Retinotopic, Not Spatiotopic, Object Location. *Cerebral Cortex*, 22(12), 2794–2810.
<https://doi.org/10.1093/cercor/bhr357>
- Hayhoe, M. M., Bensinger, D. G., & Ballard, D. H. (1998). Task constraints in visual working memory. *Vision Research*, 38(1), 125–137. [https://doi.org/10.1016/s0042-6989\(97\)00116-8](https://doi.org/10.1016/s0042-6989(97)00116-8)
- Henderson, J. M., & Hollingworth, A. (2003). Eye movements and visual memory: Detecting changes to saccade targets in scenes. *Perception & Psychophysics*, 65(1), 58–71.
<https://doi.org/10.3758/BF03194783>
- Hutchinson, J. B., Uncapher, M. R., & Wagner, A. D. (2009). Posterior parietal cortex and episodic retrieval: Convergent and divergent effects of attention and memory. *Learning & Memory (Cold Spring Harbor, N.Y.)*, 16(6), 343–356.
<https://doi.org/10.1101/lm.919109>
- Kerckhoff, G. (2001). Spatial hemineglect in humans. *Progress in Neurobiology*, 63(1), 1–27.
[https://doi.org/10.1016/s0301-0082\(00\)00028-9](https://doi.org/10.1016/s0301-0082(00)00028-9)
- Linde-Domingo, J., & Spitzer, B. (2024). Geometry of visuospatial working memory information in miniature gaze patterns. *Nature Human Behaviour*, 8(2), 336–348.
<https://doi.org/10.1038/s41562-023-01737-z>
- Liu, B., Nobre, A. C., & van Ede, F. (2022). Functional but not obligatory link between microsaccades and neural modulation by covert spatial attention. *Nature Communications*, 13(1), Article 1. <https://doi.org/10.1038/s41467-022-31217-3>
- Liu, B., Nobre, A. C., & van Ede, F. (2023). Microsaccades transiently lateralise EEG alpha activity. *Progress in Neurobiology*, 224, 102433.
<https://doi.org/10.1016/j.pneurobio.2023.102433>
- Moraesku, S., & Vlcek, K. (2020). The use of egocentric and allocentric reference frames in static and dynamic conditions in humans. *Physiological Research*, 69(5), 787–801.
<https://doi.org/10.33549/physiolres.934528>
- Nobre, A. C., Coull, J. T., Maquet, P., Frith, C. D., Vandenberghe, R., & Mesulam, M. M. (2004). Orienting Attention to Locations in Perceptual Versus Mental Representations. *Journal of Cognitive Neuroscience*, 16(3), 363–373.
<https://doi.org/10.1162/089892904322926700>

- Nobre, A. C., Gitelman, D. R., Dias, E. C., & Mesulam, M. M. (2000). Covert visual spatial orienting and saccades: Overlapping neural systems. *NeuroImage*, *11*(3), 210–216. <https://doi.org/10.1006/nimg.2000.0539>
- Nobre, A. C., Sebestyen, G. N., Gitelman, D. R., Mesulam, M. M., Frackowiak, R. S., & Frith, C. D. (1997). Functional localization of the system for visuospatial attention using positron emission tomography. *Brain: A Journal of Neurology*, *120* (Pt 3), 515–533. <https://doi.org/10.1093/brain/120.3.515>
- Sanders, D. M. W., & Cowell, R. A. (2023). The locus of recognition memory signals in human cortex depends on the complexity of the memory representations. *Cerebral Cortex*, *33*(17), 9835–9849. <https://doi.org/10.1093/cercor/bhad248>
- Sawetsuttipan, P., Phunchongharn, P., Ounjai, K., Salazar, A., Pongsuwan, S., Intrachotoo, S., Serences, J. T., & Itthipuripat, S. (2023). Perceptual Difficulty Regulates Attentional Gain Modulations in Human Visual Cortex. *Journal of Neuroscience*, *43*(18), 3312–3330. <https://doi.org/10.1523/JNEUROSCI.0519-22.2023>
- Schneegans, S., & Bays, P. M. (2017). Neural Architecture for Feature Binding in Visual Working Memory. *The Journal of Neuroscience: The Official Journal of the Society for Neuroscience*, *37*(14), 3913–3925. <https://doi.org/10.1523/JNEUROSCI.3493-16.2017>
- Shafer-Skelton, A., & Golomb, J. D. (2018). Memory for retinotopic locations is more accurate than memory for spatiotopic locations, even for visually guided reaching. *Psychonomic Bulletin & Review*, *25*(4), 1388–1398. <https://doi.org/10.3758/s13423-017-1401-x>
- Shimi, A., & Logie, R. H. (2019). Feature binding in short-term memory and long-term learning. *Quarterly Journal of Experimental Psychology*, *72*(6), 1387–1400. <https://doi.org/10.1177/1747021818807718>
- Sreenivasan, K. K., Curtis, C. E., & D’Esposito, M. (2014). Revisiting the role of persistent neural activity during working memory. *Trends in Cognitive Sciences*, *18*(2), 82–89. <https://doi.org/10.1016/j.tics.2013.12.001>
- Summerfield, J. J., Lepsien, J., Gitelman, D. R., Mesulam, M.-M., & Nobre, A. C. (2006). Orienting attention based on long-term memory experience. *Neuron*, *49*(6), 905–916. <https://doi.org/10.1016/j.neuron.2006.01.021>
- Thom, J. L., Nobre, A. C., van Ede, F., & Draschkow, D. (2023). Heading Direction Tracks Internally Directed Selective Attention in Visual Working Memory. *Journal of Cognitive Neuroscience*, 1–13. https://doi.org/10.1162/jocn_a_01976
- van Ede, F., Chekroud, S. R., & Nobre, A. C. (2019). Human gaze tracks attentional focusing in memorized visual space. *Nature Human Behaviour*, *3*(5), 462–470. <https://doi.org/10.1038/s41562-019-0549-y>
- Wallis, G., Stokes, M., Cousijn, H., Woolrich, M., & Nobre, A. C. (2015). Frontoparietal and Cingulo-opercular Networks Play Dissociable Roles in Control of Working Memory. *Journal of Cognitive Neuroscience*, *27*(10), 2019–2034. https://doi.org/10.1162/jocn_a_00838
- Watanabe, T., & Sasaki, Y. (2015). Perceptual learning: Toward a comprehensive theory. *Annual Review of Psychology*, *66*, 197–221. <https://doi.org/10.1146/annurev-psych-010814-015214>

REVIEWERS' COMMENTS

Reviewer #1 (Remarks to the Author):

This revision addressed most of my concerns well.

I appreciate that the authors collected more data and clarified several points in the manuscript, in particular those pertaining to what kind of long-term memories (LTM) this study taps into, and the issue regarding the ecological validity of the task (which was raised by several reviewers).

Overall, I remain a bit skeptical that the current set of experiments supports the rather general claim that WM and LTM processes rely on separate mechanisms. This might certainly be true in the current task, but re-reading the paper (together with the reviews) did not alleviate my concerns that this is a highly specific effect that pertains to this particular (and somewhat artificial) paradigm where only spatial strategies were tested and measured. It's long been known that in visual working memory spatial codes are particularly robust and important to maintain and retrieve information, which has not been true for LTM, where spatial codes are simply not that relevant (agreeing with Rev #3 here). The response to previous reviews confirmed that the authors also think that this study tests a very specific kind of LTM, namely "simple visual associations of object features and locations" that do not require the involvement of the "medial temporal lobe". This left me wondering whether the authors can conclude that LTM retrieval is different from WM in a more general sense that would advance theories of LTM, or whether these findings test a niche case of LTM after all. It certainly appears that the current data only allow conclusions that pertain to the SPATIAL coding between WM and LTM, and this is a much more narrow conclusion than currently put forward, and also a much more nuanced theoretical advancement.

We thank the reviewer for the thoughtful and constructive feedback. We appreciate the recognition that many concerns have been addressed with additional data and clarifications. We would like to clarify our rationale and the theoretical implications of our focus on spatial coding, as outlined below:

We acknowledge that our conclusions are necessarily constrained to the type of WM and LTM representations relevant to our task. (Indeed, this is true for the conclusions of any study employing specific types of contents and manipulations.) Nevertheless, our findings demonstrate a dissociation in the mechanisms of internal attention across WM and LTM in this particular case, by highlighting a different dependence on spatial mechanisms. The results, therefore, challenge the prevailing notion that LTM retrieval is *invariably* mediated by WM. Instead, they suggest that, under certain conditions, attentional prioritization can operate independently from WM in LTM.

We have revised the discussion section to acknowledge this limitation more clearly:

"... Further research is warranted to explore whether this dissociation extends to non-spatial or more complex forms of LTM." (page 13)

Reviewer #2 (Remarks to the Author):

The authors implemented quite extensive revisions, including an additional experiment, and these efforts have strengthened the paper in terms of narrowing down the precise impacts of, and differences between, cuing of attention in LTM vs. WM. I think the set of findings coming out of these experiments is novel and noteworthy, and will find an interested readership (though an experimental psychology journal would be a more obvious outlet in this regard). One minor remaining shortcoming is that the discussion largely re-describes the results rather than pursuing a bit more theorizing. For instance, the main finding seems to be that LTM representations of the colored shape stimuli are either disconnected from their spatial location or from the oculomotor system (or both). These observations are reported in the discussion section but not discussed very deeply in terms of why and how this might occur. For instance, perhaps the spatial information in LTM has been semanticized/translated into a verbal code? Other possibilities exist, and at least this reader would have liked to hear more informed speculation about the implications of these findings.

We appreciate the reviewer nudging us toward more speculation. This is not our natural inclination, as the question addressed by the study was *whether* (rather than *why*) internal attention in LTM was necessarily mediated through WM. Nevertheless, we have added a discussion paragraph exploring potential mechanisms underlying the dissociation between spatial/oculomotor engagement in WM versus LTM representations. We hope this addresses the request for more mechanistic speculation while grounding our interpretations in the existing literature:

"...That is, focusing within LTM facilitated sensory discrimination in a spatially specific way without accompanying spatially specific oculomotor readouts. We consider multiple possible explanations for this finding. First, previous studies have shown that oculomotor markers are strongly correlated with orienting attention within the spatial layout of WM but are not obligatory (Liu et al., 2022). The current findings might indicate that the correlation between internal spatial attention and eye movements is weaker when attention is directed to LTM contents. Furthermore, the nature of the contents stored may undergo transformations over the memory lifespan, shifting from more visual codes reflecting sensory attributes to more abstract codes including verbal or semantic dimensions (e.g., Brady et al., 2011; Hitch et al., 1995; Lifanov et al., 2021; Ranganath et al., 2004). A combination of these accounts is also possible, suggesting that focusing within LTM may rely on more flexibly activating the most relevant attributes of prior experiences to guide behavior, whereas WM representations are necessarily spatially tethered." (pages 11-12)

Reviewer #4 (Remarks to the Author):

I have carefully reviewed the revised manuscript and the authors' responses and have concluded that they have adequately addressed all the concerns. Although I remain skeptical about including Exp1, I understand that the authors prefer to retain it. Importantly, the shortcomings

of Exp1 are clearly described.

The new Experiment 3 is a valuable addition to the study. The elimination of the post-cue, as the authors noted, more directly taps into perceptual sensitivity, making the interpretations more straightforward. Moreover, the long delay between the training and testing sessions in Experiment 3 allows for ruling out priming as an alternative explanation for the observed results. This significantly strengthens the conclusions in the manuscript.

Lastly, the authors provide a detailed discussion of the possible mechanisms of internal attention in LTM, further enhancing the manuscript.

At this point, I have no further concerns and believe the manuscript is ready for publication in its current form.

We thank the reviewer for their insightful and helpful feedback. The push to include an additional experiment increasing the delay between training and testing, as well as the elimination of the post cue were important suggestions to improve the interpretability of the results. We are pleased that the reviewer finds the manuscript ready for publication and thank them for their time and expertise in helping us improve this work.

List of references mentioned in the response:

Brady, T. F., Konkle, T., & Alvarez, G. A. (2011). A review of visual memory capacity: Beyond individual items and toward structured representations. *Journal of Vision*, *11*(5), 4.

<https://doi.org/10.1167/11.5.4>

Hitch, G. J., Brandimonte, M. A., & Walker, P. (1995). Two types of representation in visual memory: Evidence from the effects of stimulus contrast on image combination. *Memory & Cognition*, *23*(2), Article 2. <https://doi.org/10.3758/BF03197217>

Lifanov, J., Linde-Domingo, J., & Wimber, M. (2021). Feature-specific reaction times reveal a semanticisation of memories over time and with repeated remembering. *Nature Communications*, *12*(1), 3177. <https://doi.org/10.1038/s41467-021-23288-5>

Liu, B., Nobre, A. C., & van Ede, F. (2022). Functional but not obligatory link between microsaccades and neural modulation by covert spatial attention. *Nature Communications*, 13(1), Article 1. <https://doi.org/10.1038/s41467-022-31217-3>

Ranganath, C., Cohen, M. X., Dam, C., & D'Esposito, M. (2004). Inferior Temporal, Prefrontal, and Hippocampal Contributions to Visual Working Memory Maintenance and Associative Memory Retrieval. *Journal of Neuroscience*, 24(16), 3917–3925.